# What If We Recaption **Billions** of Web Images with *LLaMA-3*?

Xianhang Li [* 1]  Haoqin Tu [* 1]  Mude Hui [* 1]  Zeyu Wang [* 1]  Bingchen Zhao [* 2]  Junfei Xiao [3]  Sucheng Ren [3]
Jieru Mei [3]  Qing Liu [4]  Huangjie Zheng [5]  Yuyin Zhou [1]  Cihang Xie [1]

## Abstract

Web-crawled image-text pairs are inherently noisy. Prior studies demonstrate that semantically aligning and enriching textual descriptions of these pairs can significantly enhance model training across various vision-language tasks, particularly text-to-image generation. However, large-scale investigations in this area remain predominantly closed-source. Our paper aims to bridge this community effort, leveraging the powerful and *open-sourced* LLaMA-3, a GPT-4 level LLM. Our recaptioning pipeline is simple: first, we fine-tune a LLaMA-3-8B powered LLaVA-1.5 and then employ it to recaption ~1.3 billion images from the DataComp-1B dataset. Our empirical results confirm that this enhanced dataset, Recap-DataComp-1B, offers substantial benefits in training advanced vision-language models. For discriminative models like CLIP, we observe an average of 3.1% enhanced zero-shot performance cross four cross-modal retrieval tasks using a mixed set of the original and our captions. For generative models like text-to-image Diffusion Transformers, the generated images exhibit a significant improvement in alignment with users' text instructions, especially in following complex queries. Our project page is https://www.haqtu.me/Recap-Datacomp-1B/.

## 1. Introduction

The exponential growth in data availability is one of the most paramount factors in driving the monumental successes of deep learning over the past decade (Deng et al., 2009; Lin et al., 2014; Changpinyo et al., 2021; Schuhmann et al.,

*Equal contribution  [1]University of California, Santa Cruz  [2]University of Edinburgh  [3]Johns Hopkins University  [4]Adobe Inc.  [5]University of Texas at Austin. Correspondence to: Cihang Xie <cixie@ucsc.edu>.

*Proceedings of the $42^{nd}$ International Conference on Machine Learning*, Vancouver, Canada. PMLR 267, 2025. Copyright 2025 by the author(s).

2021; Gadre et al., 2023; Fang et al., 2023). Typically, this data is sourced through web crawling with simple filtering mechanisms in place. While such an approach has facilitated large-scale data collection, exemplified by collections like LAION-400M (Schuhmann et al., 2021) and LAION-5B (Schuhmann et al., 2021) with billions of image-text records, it has inadvertently compromised data quality. As illustrated in Figure 1, these internet-crawled image-text pairs frequently exhibit misalignments between images and their corresponding textual content, and often, the textual descriptions are brief and lack detailed information.

To mitigate the noise present in web-crawled data, enhancements through post-processing—implemented via human-in-the-loop systems (Sun et al., 2023; Yu et al., 2023b) or automated pipelines (Schuhmann et al., 2021; Li et al., 2022; 2023a)—are crucial, which help to train the advanced vision-language foundation models. Notably, both the *close-sourced* DALL-E 3 (OpenAI, 2023) and SORA (OpenAI, 2024) incorporate advanced captioning techniques to relabel their training datasets, a crucial step highlighted in their technical reports. Despite various efforts to open-source and replicate these methodologies (Chen et al., 2023a; Li et al., 2022; 2023a; Liu et al., 2023b; Yu et al., 2023a; Fan et al., 2024; Rotstein et al., 2023), the community continues to face significant challenges in accessing high-quality, well-aligned image-text data at scale (*e.g.*, at the billion level) for training advanced vision-language foundation models.

This paper endeavors to contribute to this community initiative, inspired specifically by the release of LLaMA-3 (Meta LLaMA Team, 2024), a model demonstrating GPT-4-level capabilities across a variety of linguistic tasks. Additionally, recent studies have shown that leveraging LLaMA-3 can significantly enhance model performance on vision-language tasks (Liu et al., 2024; Xu et al., 2024), comparable to those achieved by GPT-4V (Achiam et al., 2023a). In response, we employ LLaMA-3 to develop our advanced captioner model. Our approach is straightforward: we first train a LLaMA-3-powered LLaVA model to act as an image captioner, which is then utilized to recaption the entire DataComp-1B dataset. As depicted in Figure 1, the resulting dataset, dubbed Recap-

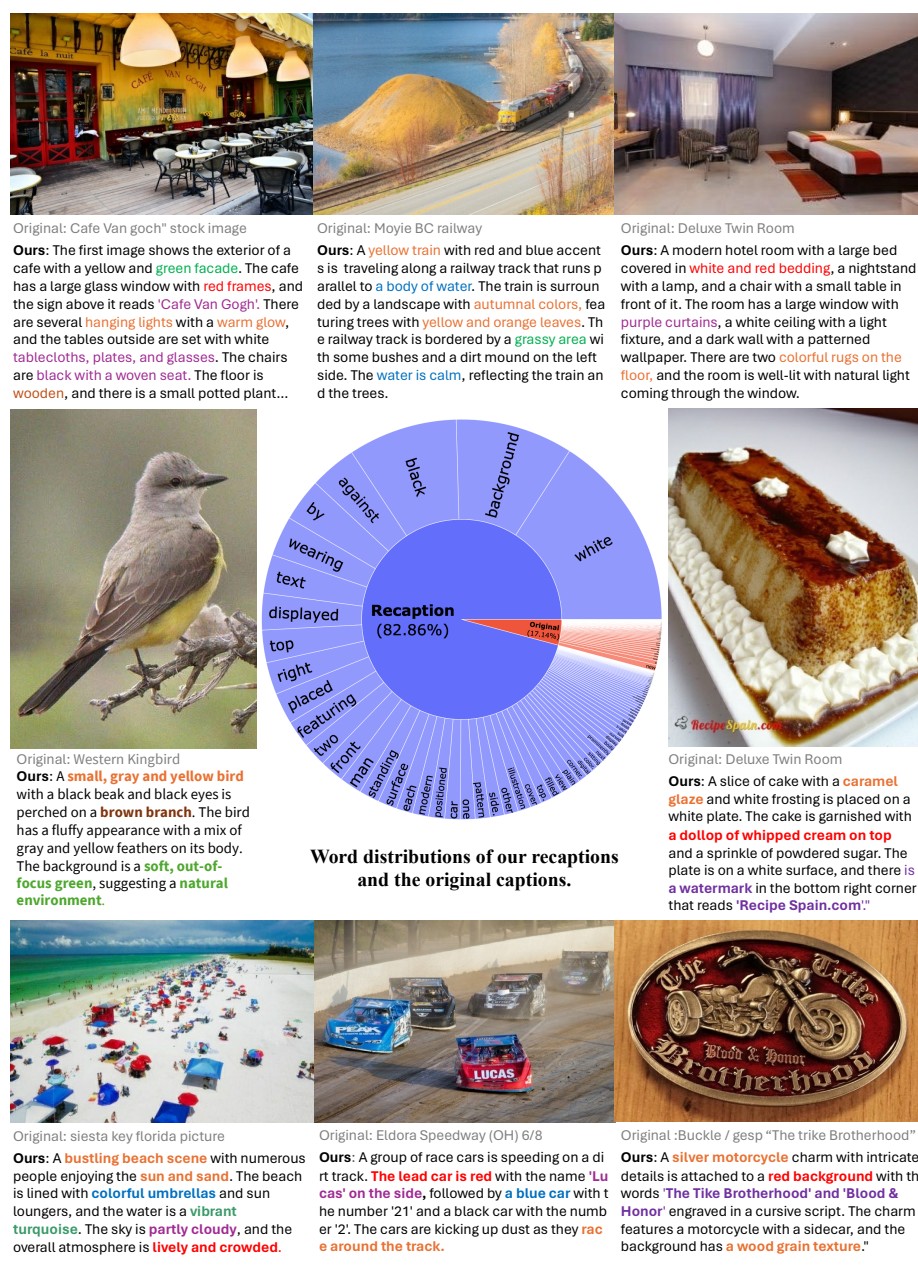

Original: Cafe Van goch" stock image
**Ours**: The first image shows the exterior of a cafe with a yellow and green facade. The cafe has a large glass window with red frames, and the sign above it reads 'Cafe Van Gogh'. There are several hanging lights with a warm glow, and the tables outside are set with white tablecloths, plates, and glasses. The chairs are black with a woven seat. The floor is wooden, and there is a small potted plant...

Original: Moyie BC railway
**Ours**: A yellow train with red and blue accents is traveling along a railway track that runs parallel to a body of water. The train is surrounded by a landscape with autumnal colors, featuring trees with yellow and orange leaves. The railway track is bordered by a grassy area with some bushes and a dirt mound on the left side. The water is calm, reflecting the train and the trees.

Original: Deluxe Twin Room
**Ours**: A modern hotel room with a large bed covered in white and red bedding, a nightstand with a lamp, and a chair with a small table in front of it. The room has a large window with purple curtains, a white ceiling with a light fixture, and a dark wall with a patterned wallpaper. There are two colorful rugs on the floor, and the room is well-lit with natural light coming through the window.

Original: Western Kingbird
**Ours**: A small, gray and yellow bird with a black beak and black eyes is perched on a brown branch. The bird has a fluffy appearance with a mix of gray and yellow feathers on its body. The background is a soft, out-of-focus green, suggesting a natural environment.

**Word distributions of our recaptions and the original captions.**

Original: Deluxe Twin Room
**Ours**: A slice of cake with a caramel glaze and white frosting is placed on a white plate. The cake is garnished with a dollop of whipped cream on top and a sprinkle of powdered sugar. The plate is on a white surface, and there is a watermark in the bottom right corner that reads 'Recipe Spain.com'."

Original: siesta key florida picture
**Ours**: A bustling beach scene with numerous people enjoying the sun and sand. The beach is lined with colorful umbrellas and sun loungers, and the water is a vibrant turquoise. The sky is partly cloudy, and the overall atmosphere is lively and crowded.

Original: Eldora Speedway (OH) 6/8
**Ours**: A group of race cars is speeding on a dirt track. The lead car is red with the name 'Lucas' on the side, followed by a blue car with the number '21' and a black car with the number '2'. The cars are kicking up dust as they race around the track.

Original :Buckle / gesp "The trike Brotherhood"
**Ours**: A silver motorcycle charm with intricate details is attached to a red background with the words 'The Tike Brotherhood' and 'Blood & Honor' engraved in a cursive script. The charm features a motorcycle with a sidecar, and the background has a wood grain texture."

*Figure 1.* Examples of the original caption and our recaption in DataComp-1B, and word distributions. The word distribution compares the frequency of word usage in the re-captioned data to that in the original captions.

DataComp-1B, features enhanced textual descriptions and improved alignment with corresponding images, clearly surpassing its web-crawled counterparts. These quality enhancements are further quantitatively verified in Section 4.

Comprehensive evaluations highlight the significant improvements that Recap-DataComp-1B contributes to the training of advanced vision-language foundation models. Notably, this dataset enables CLIP models to achieve significant enhancements in their zero-shot cross-modal retrieval

capabilities (*e.g.*, 64.8% > 61.7% over four cross-modal retrieval tasks). It also improves the alignment between generated images and text instructions in text-to-image generative models pre-trained on our dataset, resulting in higher quality images and better relevance to the input—demonstrated by an 8.4 lower FID and a 3.1% higher CLIP score. We hope that the release of Recap-DataComp-1B will catalyze further developments in advanced vision-language foundation models, particularly encouraging the development within the open-source community.

## 2. Related works

**Vision-Language Foundation Models.** CLIP (Radford et al., 2021a) is one of the pioneering foundation models to connect image and text. By training on millions, and even billions, of image-text pairs (Changpinyo et al., 2021; Desai et al., 2021; Fang et al., 2023; Gadre et al., 2023; Schuhmann et al., 2022b; 2021; Sharma et al., 2018; Srinivasan et al., 2021), CLIP markedly showcases excessively strong zero-shot capacities, and furthermore, lays the cornerstone for building more advanced vision-language foundation models (Alayrac et al., 2022; Li et al., 2022; 2023a; Wang et al., 2022; Liu et al., 2023b; 2024; Chen et al., 2023b; Bai et al., 2023; Xu et al., 2024). Apart from discriminative vision-language understanding, text-to-image generation models (Ding et al., 2021; Nichol et al., 2021; OpenAI, 2023; Peebles & Xie, 2023; Ramesh et al., 2022; 2021; Rombach et al., 2021; Saharia et al., 2022; Yu et al., 2022a) have transformed the field of AI-generated content, facilitating the creation of high-quality images from natural language descriptions.

**Enhancing Image-Text Data.** Web-crawled image-text data (Schuhmann et al., 2021; Gadre et al., 2023; Fang et al., 2023) commonly face the problems of image-text misalignment and the low-quality of textual descriptions. Typically, there are two popular ways for improving the quality of these image-text pairs: 1) *data filtering* removes misaligned image-text pairs using various methods such as cleaning strategies (Schuhmann et al., 2022b; Gadre et al., 2023; Xu et al., 2023), pretrained models (Li et al., 2022; Schuhmann et al., 2021; Gadre et al., 2023), and human-assisted systems (Sun et al., 2023; Yu et al., 2023b; Zhang et al., 2023); 2) *data recaptioning* improves the textual quality of image-text pair via generating new captions, which is the focus of this paper. To recaption data, LaCLIP (Fan et al., 2024) utilizes large language models (LLMs) like ChatGPT to rewrite the original captions; Nguyen *et al.* (Nguyen et al., 2024) employ BLIP2 (Li et al., 2023a) to recaption images. More recently, advanced large multimodal models have been applied to further enhance the quality of image captioning. For example, ShareGPT4V (Chen et al., 2023a) employs GPT-4V to create highly descriptive captions from carefully crafted prompts and corresponding image inputs; the resulting dataset has significantly benefited the training of various models (Chen et al., 2024a; Zhang et al., 2024a; Chu et al., 2024; Lin et al., 2024; Fei et al., 2024; Awadalla et al., 2024). However, scaling such prompting with GPT-4V to billions of records is less practical, as it will drastically increase the monetary cost (of intensively calling OpenAI APIs) by more than 10,000×.

Our paper mostly follows the approach presented in (Chen et al., 2024b; Lu et al., 2023; Zhang et al., 2024a; Chu et al., 2024; Huang et al., 2024), where advanced open-source mul-

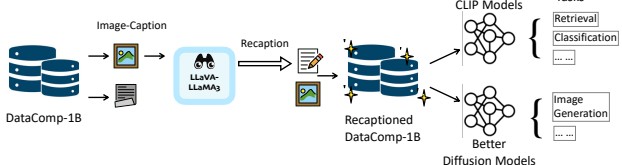

*Figure 2.* The illustration of our recaptioning pipeline on DataComp-1B. We use LLaMA-3-powered LLaVA to reception images, which enables us to train stronger CLIP models and Text-to-Image Diffusion models.

timodal models like LLaVA (Liu et al., 2023b) are employed for recaptioning purposes. However, our approach is distinguished by two major aspects: 1) we strongly enhance the LLM module in LLaVA, *i.e.*, building with LLaMA-3; and 2) our recaptioning efforts are executed on a billion-scale dataset.

## 3. Recaptioning Pipeline

Our recaptioning pipeline is centered around the advanced LLM LLaMA-3 (Meta LLaMA Team, 2024), which achieves exceptionally strong performance in language understanding, reasoning, code generation, math problems, *etc*. (Chiang et al., 2024; Stevens, 2024). Specifically, we utilize the LLaVA framework (Liu et al., 2023b) to fully harness its capabilities for visual understanding. We describe the detailed training procedures below.

### 3.1. Model details

**Model Configuration.** We follow the setup of LLaVA-1.5 (Liu et al., 2023a) to build our captioner model, except that we use LLaMA-3-8B as the language decoder because of its superior performance. The visual branch of CLIP ViT-L/14 (Radford et al., 2021b) is used as the vision encoder. Two trainable MLP layers are employed on top of the vision encoder to project visual features into the language embedding space.

**2-Stage Training.** We also follow LLaVA-1.5 (Liu et al., 2023a) for model training. Essentially we conduct instruction-tuning on the pre-trained LLM with its original auto-regressive training objective. In the first stage, only the projection MLP is trained; in the second stage, we fine-tune both the projection MLP and the language decoder. Note that the vision encoder remains frozen all the time. Following the protocols in LLaVA (Liu et al., 2023a), 558k image-text pairings filtered from LAION (Schuhmann et al., 2022b), CC (Changpinyo et al., 2021),and SBU (Ordonez et al., 2011) are used as training data in the first stage; then 665k instructions-following data from LLaVA-1.5 (Liu et al., 2023a), containing image-grounded conversation, image descriptions, and image-based complex reasoning tasks, are used for the second stage of training. *To help our model*

*generate higher-quality captions, we use the image-text pairs from HQ-Edit dataset (Hui et al., 2024) for further tuning, which distinguishes our approach from concurrent work (Hinck et al.).*

Table 1. Performance comparison of LLaVA.

| Model | LLaVA-1.5-7B | LLaVA-1.5-13B | LLaVA-1.5-LLaMA3-8B (ours) | GPT-4V |
|-------|--------------|---------------|----------------------------|--------|
| MMMU | 33.6 | 36.4 | **37.5** | 56.8 |
| MM-Vet | 33.9 | 36.3 | **36.5** | 44.6 |

**Evaluations.** To probe the visual understanding and reasoning ability of our LLaVA-1.5-LLaMA3-8B model, we opt for two comprehensive multi-modal evaluation benchmarks, MMMU (Yue et al., 2024) and MM-Vet (Yu et al., 2024). These benchmarks assess a broad range of capabilities such as recognition, spatial awareness, OCR, knowledge, and language generation. As reported in Table 1, on both benchmarks, our LLaVA-1.5-LLaMA3-8B model surpasses the vanilla LLaVA-1.5-7B model by a significant margin. These results also match, or even outperform, the considerably larger LLaVA-1.5-13B model, demonstrating the superior visual understanding and reasoning ability of our model.

### 3.2. Recaptioning DataComp-1B

With this advanced LLaVA model, we next use it to generate captions in a scalable and detailed manner, given the visual input, and the following text prompt: *"Please generate a detailed caption of this image. Please be as descriptive as possible."* As for the dataset, we opt for DataComp-1B (Gadre et al., 2023), a widely accessible, large-scale vision-language dataset comprising ~1.3 billion web-crawled image-text pairs. To ensure its quality, DataComp-1B is already a curated subset from a much larger collection of 12.8 billion image-text pairs and has been subjected to rigorous preprocessing which includes safety checks, deduplication, CLIP score filtering, and image-based filtering. Despite these efforts, the quality of the original captions in DataComp-1B still exhibits relatively low quality.

We apply our well-trained LLaVA-1.5-LLaMA3-8B model to recaption the entire DataComp-1B dataset. Specifically, captions are generated auto-regressively via greedy decoding, with the maximum output token length set to 128. Details on the computation cost are provided in Appendix B. We term this newly recaptioned dataset *Recap-DataComp-1B*.

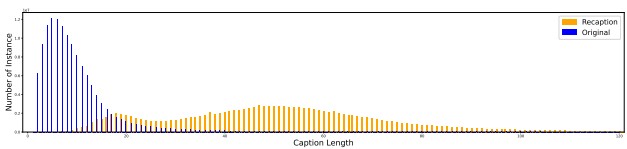

*Figure 3.* Instance length distributions from ~133 million sampled examples of both the original captions and our recaptioned data in DataComp-1B.

## 4. Analyzing Recap-DataComp-1B

This section collects and presents a quantitative analysis of our generated captions on DataComp-1B. We primarily focus on two aspects: 1) the inherent features of the captions, including word distributions and average lengths; and 2) the semantic quality of the captions, evaluated in terms of the matching similarity between images and captions and the inherent textual quality of the captions.

### 4.1. Word & Length Distribution

We begin our analysis by comparing the word frequency distributions between our recaptioned content and that from the original DataComp-1B, as illustrated in Figure 1, analyzing a randomly sampled subset of approximately 0.35 billion instances. Our findings reveal that the recaptioned content displays a considerably richer vocabulary, capturing 82.86% tokens of the word collections from both ours and the original caption data. Additionally, there is a noticeable variety in the usage of nouns and adjectives in our captions (*e.g.*, "white" and "background"). We argue that this increased lexical diversity is a direct consequence of the extended length of our data. We thus present the distribution of instance caption lengths in Figure 3 to highlight this difference. On average, our recaptioned data demonstrates a longer sequence length of 49.43, whereas the original DataComp captions have a much shorter length of 10.22. These observations validate that our Recap-DataComp-1B surpasses the original DataComp-1B version in terms of both caption length and diversity.

> **GPT-4V Evaluation Instruction:**
> [Image Caption]
> CAPTION
>
> Rate whether the caption is of high-quality and fluent and correctly matches the given image. The rating should be 1-5, where 1 is incorrect and not fluent at all, and 5 is correct and very fluent. Try to just give a numerical rating.
>
> Your response should be in the format:
> Rating: (int)

### 4.2. GPT-4V & CLIP Evaluations

Next, we evaluate the semantic quality of recaptioned content using two models: 1) CLIP (Radford et al., 2021a), which measures the semantic similarity between captions and images, and 2) GPT-4V (Achiam et al., 2023b), which assesses the fluency and alignment of captions with the given images.

For the CLIP evaluation, we analyze a subset of 180,000 image-text pairs. Interestingly, we note that, when using the

standard CLIP-large model with ~428M parameters for this measurement, our recaptioned content performs just comparably to the original captions (49.57 vs. 50.43). We attribute this result primarily to the limitations of the standard CLIP model, which is trained on 'short' captions and may inadequately capture the nuances in semantic similarity for longer captions. To probe deeper into semantic alignment between long captions and images, we utilize the LongCLIP-Large model (Zhang et al., 2024a), which is specifically fine-tuned to handle longer captions. With this setup, the LongCLIP score of our newly generated caption impressively attains 89.91, nearly $9\times$ higher than the LongCLIP score of the original DataComp captions (i.e., only 10.09).

In addition, to evaluate both the textual quality and the alignment of the captions with their corresponding images, we randomly select 10,000 instances for GPT-4V (Achiam et al., 2023b) evaluation, employing the prompting strategy outlined above (CAPTION is the textual input), as per (Padlewski et al., 2024; Lee et al., 2024). We can observe that our recaptioned content achieves markedly superior ratings, registering an average rating increase of 0.43 (from 3.71 to 4.14). Together with the findings from Section 4.1, this confirms the superior quality of our newly generated captions, in terms of length, vocabulary diversity, semantics, and image-text alignment.

## 5. Training CLIPs with Recaptions

CLIP (Radford et al., 2021a) stands as a widely utilized vision-language model, where an image encoder and a text encoder are jointly trained to predict correct matches across entire batches of image-text pairs. In this section, we delve into the advantages of training CLIP models with our Recap-DataComp-1B dataset. We anticipate that CLIP models trained on this dataset will exhibit superior zero-shot cross-modal retrieval capabilities and enhanced text understanding, especially with long and complex textual inputs, given the improved quality of our recaptions.

### 5.1. Experiment settings

**Training.** For reference, we term the CLIP model trained on our Recap-DataComp-1B dataset as Recap-CLIP. Our training pipeline primarily follows CLIPA (Li et al., 2023b;c), which incorporates a two-state training, i.e., a pre-training process with a small image size followed by a fine-tuning stage incorporating a larger image resolution. We set the text token length to 128 to accommodate the learning of long captions presented in Recap-DataComp-1B. We conduct experiments using three model scales: S/16, B/16, and L/16, with details listed in Appendix Table 7. The AdamW (Loshchilov & Hutter, 2017) optimizer is used for training. In the pre-training phase, the model is trained with 2.56 billion samples with a reduced image size of 112,

including a warm-up phase involving 51.2 million samples. The batch size and base learning rate are set to 32,768 and 8e-6, respectively. For the subsequent fine-tuning phase, we increase the image size to 224 and train the model on 128 million samples with a 25.6 million sample warm-up. We adjust the batch size to 16,384 and the learning rate to 4e-7.

**Evaluation.** The efficacy of Recap-CLIP is gauged via several metrics. We evaluate zero-shot image classification on the ImageNet-1K dataset (Russakovsky et al., 2015) and assess zero-shot cross-modal retrieval performance using the validation set of MSCOCO 2014 (Lin et al., 2014) and the test set of Flickr30K (Young et al., 2014)[1], following the established practices (Radford et al., 2021a; Li et al., 2023b; Zhai et al., 2023; 2022).

We present our results from three aspects. First, we explore the impacts of differing mix ratios between original captions and our enhanced recaptions on CLIP performance. Next, we analyze the effects of enlarging the size of the CLIP text encoder. Lastly, we investigate the text understanding capability of our Recap-CLIP, via testing on VG-Attribute (Yuksekgonul et al., 2022), which evaluates attributes understanding ability, and Urban1K (Zhang et al., 2024a), which tests the model's ability to handle long text.

### 5.2. Training with Mixed Captions

As pointed out by DALL-E 3 (OpenAI, 2023), blending both the briefgenuine captions and the long informative generated captions can effectively prevent the model from unwanted overfitting to recaption data. Therefore, we hereby first study the effect of varying mix ratios between the original captions and our recaptions on the training of the Recap-CLIP B/16 model, as detailed in Table 7. Specifically, for each sample in a training batch, we randomly sample the original caption with probability $0 \leq p \leq 1$ and our captions with probability $1 - p$, referring to the mixed ratio:

$$\text{Caption} = \begin{cases} \text{Original} & \text{with probability } p \\ \text{Recaption} & \text{with probability } 1 - p \end{cases}$$

This strategy ensures that each batch contains a mixture of our recaption and the original captions controlled by probability $p$. The randomness allows each sample to encounter different captions across training epochs, potentially enhancing the model's generalization.

**Main results.** Our findings are summarized in Table 3. We observe that reducing the mixed ratio $p$ (i.e., increasing the proportion of our recaption data) initially leads to an improvement followed by a decline in cross-modal retrieval performance. This initial improvement suggests that high-quality recaptioned data effectively enhances contrastive

---

[1]We employ the widely used Karpathy split (Karpathy & Fei-Fei, 2015) of MSCOCO and Flickr30K.

*Table 2.* **Train with larger text encoder.** We set $p = 0.8$ for recaption-based models. We report zero-shot top-1 accuracy on ImageNet-1K and top-1 recall on COCO and Flickr30K.

| vision encoder | text encoder | re-caption | ImageNet-1K | COCO R@1 | | Flickr30K R@1 | |
|---|---|---|---|---|---|---|---|
| | | | Validation | I→T | T→I | I→T | T→I |
| S/16 | small | ✗ | 61.6 | 50.9 | 30.5 | 75.3 | 54.5 |
| | small | ✔ | 61.1 $_{-0.5\%}$ | 55.8 $_{+4.9\%}$ | 35.0 $_{+4.5\%}$ | 79.6 $_{+4.3\%}$ | 59.2 $_{+4.7\%}$ |
| | base | ✗ | 62.7 | 53.0 | 32.2 | 78.7 | 57.3 |
| | base | ✔ | 62.1 $_{-0.6\%}$ | 56.7 $_{+3.7\%}$ | 36.0 $_{+3.8\%}$ | 80.4 $_{+1.7\%}$ | 60.3 $_{+3.0\%}$ |
| B/16 | base | ✗ | 70.5 | 59.5 | 38.9 | 84.1 | 64.4 |
| | base | ✔ | 69.8 $_{-0.7\%}$ | 62.8 $_{+3.3\%}$ | 42.7 $_{+3.8\%}$ | 86.7 $_{+2.6\%}$ | 67.1 $_{+2.7\%}$ |
| | large | ✗ | 71.1 | 61.0 | 40.2 | 86.8 | 67.6 |
| | large | ✔ | 70.5 $_{-0.6\%}$ | 64.5 $_{+3.5\%}$ | 43.4 $_{+3.2\%}$ | 88.3 $_{+1.5\%}$ | 68.9 $_{+1.3\%}$ |
| L/16 | large | ✗ | 76.5 | 62.5 | 44.2 | 89.0 | 71.1 |
| | large | ✔ | 76.1 $_{-0.4\%}$ | 66.8 $_{+4.3\%}$ | 47.8 $_{+3.6\%}$ | 91.3 $_{+2.3\%}$ | 73.9 $_{+2.8\%}$ |
| | huge | ✗ | 76.9 | 65.2 | 46.0 | 90.5 | 73.2 |
| | huge | ✔ | 76.3 $_{-0.6\%}$ | 68.5 $_{+3.3\%}$ | 49.1 $_{+3.1\%}$ | 91.0 $_{+0.5\%}$ | 75.6 $_{+2.4\%}$ |

learning. However, the subsequent decrease indicates that the original captions from DataComp-1B provide necessary training regularization, preventing the model from overly adapting to the specific qualities of the recaption data. Interestingly, we also observe that the performance of CLIP is relatively insensitive to certain variations in the mix ratio $p$, as evidenced by the consistent enhancement over the baseline (*i.e.* $p$=1.0) across all four different cross-modal retrieval metrics when varying $p$ from 0.2 (80% recaption data) to 0.9 (10% recaption data). For instance, setting $p$ at 0.9 and 0.2 both yields a similar performance enhancement of ∼2.7%, with the peak performance occurring at $p$=0.4, which delivers a substantial ∼5% boost.

But meanwhile, we note that incorporating our recaptions (negatively) affects the zero-shot classification task, exemplified by the consistent performance degradation across varying $p$ values from 0 to 0.9. The phenomenon is similarly observed in the recent work (Zhang et al., 2024a) where they note directly fine-tuning on long text can significantly hurt the CLIP performance. Several works (Zheng et al., 2024; Liu et al., 2023d) propose novel techniques for enhancing learning with long texts. In this study, given our primary focus is on assessing the quality of Recap-DataComp-1B, we choose the ratio $p = 0.8$ to strike a promising balance between the classification performance (*i.e.*, only marginally drops 0.7%) and the cross-modal retrieval performance (*i.e.*, with a significant 3.1% boost on average) for later ablations.

### 5.3. Training with Larger Text Encoder

We hereby investigate how the size of the text encoder affects models trained on a mixture of the original captions and our recaptions (with $p = 0.8$). Specifically, we keep the configuration of the vision branch as in appendix Table 7 and only twitch the text encoder. For instance, in the case of the S/16 model, we change from a smaller text encoder with 33M parameters to a larger, base-sized one with 53M

*Table 3.* **Train with mixed captions. We choose Recap-CLIP-B/16 for this ablation.** Larger $p$ represents a higher ratio of the original caption. We report top-1 zero-shot classification accuracy on ImageNet-1K and top-1 recall for retrieval tasks.

| mixed ratio $p$ | ImageNet-1K | COCO R@1 | | Flickr30K R@1 | |
|---|---|---|---|---|---|
| | Validation | I→T | T→I | I→T | T→I |
| 0.0 | 36.0 | 53.0 | 34.1 | 74.1 | 53.5 |
| 0.1 | 59.6 | 62.5 | 41.6 | 84.2 | 65.5 |
| 0.2 | 63.8 | 61.7 | 42.4 | 86.8 | 67.0 |
| 0.3 | 65.5 | 62.7 | 42.6 | 86.2 | 68.4 |
| 0.4 | 66.7 | 63.4 | 43.2 | **87.6** | **68.2** |
| 0.5 | 67.9 | 62.3 | **43.3** | 85.6 | 67.7 |
| 0.6 | 68.8 | **63.6** | 43.1 | 86.2 | 68.2 |
| 0.7 | 69.0 | 62.9 | 42.8 | 85.7 | 68.1 |
| 0.8 | 69.8 | 62.8 | 42.7 | 86.7 | 67.1 |
| 0.9 | 70.3 | 62.8 | 41.8 | 86.1 | 66.8 |
| 1.0 | **70.5** | 59.5 | 38.9 | 84.1 | 64.4 |

parameters.

**Main Results** Our results, as shown in Table 2, first, demonstrate the effectiveness of using enhanced captions compared to original ones. On average, retrieval tasks show improvements of 4.6%, 3.1%, and 3.3% for small, base, and large models, respectively. Second, enlarging the text encoder further boosts performance across all model scales. When the text encoder is enlarged, re-captioning consistently delivers significant improvements, suggesting that enhanced captions provide benefits across models of varying scales.

### 5.4. More evaluations on text understanding

Recent works demonstrate that CLIP models suffer from poor long context understanding and delicate attribute understanding (Yuksekgonul et al., 2022; Zhang et al., 2024a). Given the long, enriched, and better-aligned captions, we expect Recap-CLIP to exhibit better text understanding capability. Thus, we evaluate our Recap-CLIP model on two benchmarks: (1) Urban1K (Zhang et al., 2024a), a long-caption image-text retrieval benchmark that contains 1k urban images and corresponding GPT-4V captions; (2) VG-Attribution (Yuksekgonul et al., 2022), a modified version of

*Table 4.* **Comparison with other CLIP models** trained on public or private dataset. We report top-1 ImageNet-1K classification accuracy and recall of image and text retrieval on COCO and Flickr30K.

| method | model size | # patches | dataset | public | IN-1K val. | Flickr30K R@1 T→I | Flickr30K R@1 I→T | COCO R@1 T→I | COCO R@1 I→T |
|---|---|---|---|---|---|---|---|---|---|
| CLIP (Radford et al., 2021a) | Large | 256 | WIT-400M (Radford et al., 2021a) | ✗ | 75.5 | 65.0 | 85.2 | 36.5 | 56.3 |
| CLIP (Gadre et al., 2023) | Large | 256 | DataComp-1B (Gadre et al., 2023) | ✔ | 79.2 | 73.4 | 89.0 | 45.7 | 63.3 |
| OpenCLIP (Ilharco et al., 2021) | Large | 256 | LAION-2B (Schuhmann et al., 2022a) | ✔ | 75.5 | 75.5 | 89.5 | 46.5 | 63.4 |
| SigLIP (Zhai et al., 2023) | Large | 256 | WebLI-5B (Chen & Wang, 2022) | ✗ | **80.5** | 79.0 | 91.8 | 52.3 | 70.8 |
| Recap-CLIP | Large | 256 | Recap-DataComp-1B | ✔ | 79.3 | **79.5** | **94.1** | **53.7** | **72.0** |
| CLIP (Fang et al., 2023) | Huge | 729 | DFN-5B (Fang et al., 2023) | ✗ | **84.4** | 82.0 | 94.0 | 55.6 | 71.9 |
| SigLIP (Zhai et al., 2023) | SO(400M) | 729 | WebLI-5B (Chen & Wang, 2022) | ✗ | 83.1 | **83.0** | 94.3 | 54.2 | 72.4 |
| Recap-CLIP | Huge | 256 | Recap-DataComp-1B | ✔ | 81.0 | 81.3 | **94.8** | **54.5** | **73.1** |

*Table 5.* Comparison on the Urban-1K and VG-Attribute.

| method | re-caption | Urban-1K I→T | Urban-1K T→I | VG Attribute |
|---|---|---|---|---|
| OpenAI-CLIP-B/16 (Radford et al., 2021a) | ✗ | 67.4 | 53.3 | 62.6 |
| OpenCLIP-B/16 (Ilharco et al., 2021) | ✗ | 62.5 | 63.1 | 59.9 |
| Recap-CLIP-B/16 | ✗ | 53.2 | 50.9 | 57.1 |
| Recap-CLIP-B/16 | ✔ | 85.0 +31.8% | 87.3 +36.4% | 66.4 +9.1% |
| Recap-CLIP-L/16 | ✗ | 69.8 | 64.6 | 60.1 |
| Recap-CLIP-L/16 | ✔ | 89.0 +19.2% | 91.8 +27.2% | 66.8 +6.7% |

Visual Genome (Krishna et al., 2017) to test model abilities to attribute properties to objects, as shown in Tab. 5.

We observe consistent significant improvement if the model is trained on our Recap-Datacomp-1B dataset. For both text-to-image and image-to-text retrieval on Urban-1K dataset, our Recap-CLIP models surpass the vanilla baseline by at least 19% and sometimes up to an astonishingly high 36%. On the VG-attribution dataset, it is worth noting that our Recap-CLIP brings a performance boost very close to that of the NegCLIP fine-tuning (Yuksekgonul et al., 2022) (*e.g.* ∼9% *vs.* 10%), a lightweight downstream fine-tuning process designed to boost CLIP ability to understand attribute and order. Nonetheless, it is noteworthy that our Recap-CLIP is naturally equipped with better text understanding ability, without any specific targeted fine-tuning, indicating the importance of better captions in web-scale data.

### 5.5. Scaling-up Recap-CLIP

To compare with current state-of-the-art CLIP models, by utilizing the training recipe discussed above ($p = 0.8$ and a larger text encoder), we simply scale up the training schedule by 5 ×, processing 12.8 billion samples, following common practice in training CLIP (Radford et al., 2021a; Li et al., 2023c; Ilharco et al., 2021). The hyperparameters like batch size learning rate mainly follow CLIPA (Li et al., 2023b), and we train the L/14 and H/14 models. The results are shown in Table 4. First, when comparing advanced CLIP models trained on publicly available datasets like LAION-2B (Schuhmann et al., 2022a) and DataComp-1B (Gadre et al., 2023), our Recap-CLIP L/14 model achieves the best performance on zero-shot ImageNet-1K classification and COCO/Flickr30K retrieval tasks with a notable mar-

gin. Specifically, compared to the original DataComp-1B L/14 model, our enhanced dataset and training result in significant performance improvements, with average gains of **5.6%** and **8.4%** on Flickr and COCO. Second, compared with the current state-of-the-art models trained on private datasets like WebLI-5B (Chen & Wang, 2022), our model demonstrates much higher training efficiency. For instance, SigLIP (Zhai et al., 2023) is trained on 45 billion samples from a 5 billion-image in-house dataset, while our model, using only 12.8 billion training samples and significantly fewer model flops, achieves a better retrieval performance. These results clearly highlight the quality of our proposed dataset and training method at scale.

## 6. Training Text-to-Image Models with Recaptions

It has been known to the research community that training with generated (high-quality) pseudo-captions improves text-to-image generative models in terms of generation quality and prompt following ability (Chen et al., 2024b;a; Betker et al., 2023), primarily due to the low information and high noise density presented in the original web-crawled captions. Therefore, we evaluate the quality of our generated captions by training Text-to-Image (T2I) generative models on Recap-DataComp-1B for further justification. We expect the enriched information in the generated descriptions to better align the visual content in images, and thus improve the performance of the T2I models.

**Training.** We adopt Diffusion Transformers (DiT) (Peebles & Xie, 2023) as our T2I model, where the text condition is extracted with a CLIP text encoder (Radford et al., 2021a), and then injected into each DiT block with the cross-attention.We employ the original CLIP model to establish a consistent evaluation baseline that aligns with community standards, enabling clearer analysis of the relationship between data modifications and performance changes. Specifically, we follow the image preprocessing pipeline in DiT (Peebles & Xie, 2023), where the images are preprocessed to have a square resolution of 256. The model

*Table 6.* Text-to-Image evaluation on COCO-30K results of DiT-BASE/4, trained with different mix ratios on Recap-DataComp-1B. Note for GPT-4V Score, we use a subset of 3K for the evaluation.

| Training | Evaluation | | | | | |
|---|---|---|---|---|---|---|
| | Raw | | Our COCO-Recap | | | |
| mixed ratio $p$ | FID↓ | CLIP Score↑ | FID↓ | CLIP Score↑ | Recap-Clip Score↑ | GPT-4V Score↑ |
| 0.00 | 37.6 | 29.2 | $27.8_{-8.4}$ | $32.5_{+3.1\%}$ | $\mathbf{28.3}_{+8.4\%}$ | $\mathbf{2.53}_{+1.1}$ |
| 0.05 | 38.5 | 29.1 | 27.9 | 32.5 | 28.0 | 2.51 |
| 0.10 | 36.0 | 29.7 | **27.2** | 32.7 | 28.2 | 2.51 |
| 0.15 | 35.8 | 29.9 | 28.2 | **33.0** | 28.1 | 2.45 |
| 0.20 | 35.8 | 29.8 | 28.4 | 32.7 | 28.0 | **2.53** |
| 0.50 | 35.3 | 29.3 | 30.2 | 31.9 | 26.7 | 2.13 |
| 0.75 | 31.3 | 29.4 | 32.7 | 31.2 | 25.8 | 1.89 |
| 1.00 | 32.5 | 28.9 | 36.2 | 29.3 | 19.9 | 1.40 |

is trained on visual latent extracted using a pretrained autoencoder with a downsampling ratio of 8 (Rombach et al., 2021). Similar to the setup in previous experiments, the training text consists of a mixture of raw captions from Datacomp-1B, with a specified proportion $p$, and the rest of the captions replaced by refined captions from Recap-Datacomp-1B. Moreover, the training batch size is 2048, and the AdamW optimizer (Loshchilov & Hutter, 2017) is used with a constant 1e-4 learning rate, without any warm-up schedule or weight decay. We name the resulting model Recap-DiT.

**Evaluation.** For sampling, we set the classifier-free guidance scale as 10 and use 250 DDPM steps to generate 30k images with captions from MSCOCO and our improved generated captions for zero-shot generation evaluation. Raw means evaluating with original coco captions, Our COCO-Recap means evaluating with recapted COCO captions. We calculate Fréchet Inception Distance (FID) (Heusel et al., 2017) with the reference images from MSCOCO (Lin et al., 2014) and CLIP score with both OpenAI ViT-B/32 model (Radford et al., 2021a) and our own Recap-CLIP ViT-L/16 model, following the established pipeline in prior T2I works (Betker et al., 2023; Yu et al., 2022b; Sauer et al., 2023a; Kang et al., 2023; Liu et al., 2023c; Zhou et al., 2024; Sauer et al., 2023b). Additionally, following the GPT-4V metric introduced in Section 4.2, we randomly select a subset of 3,000 our generated images for GPT-4V evaluation.

**Main results.** We report our observations in Tab. 6. Interestingly, when using raw COCO captions to generate 30,000 images for evaluation, the model trained with data integrated with our Recap-Datacomp (for $p < 1$) demonstrates a better CLIP score, indicating improved vision-language alignment. However, there is no significant improvement observed in terms of FID. Our hypothesis is that the model adapts to the more informative and descriptive prompts, and could unleash its full potential only when similar informative testing prompts are provided.

Therefore, in another setting, we evaluate images generated using our LLaVA-1.5-LLaMA3-8B recaptioned version of

the raw COCO captions. Here, we observe consistent and significant improvements in both FID and CLIP scores, particularly when more than half of the recaptioned data are integrated into the training dataset. Notably, models trained on Recap-Datacomp-1B ($p = 0$) surpass those trained on the vanilla Datacomp-1B ($p = 1$) by a large margin, with improvements observed in FID (-8.4), CLIP score (+3.1), Recap-CLIP score (+8.4), and GPT-4V score (+1.1). These observations justify that Recap-Datacomp-1B better reveals the potential of text-to-image models in generating images with high visual quality and improved alignment with textual conditions. The comparison between p = 1.0 and p = 0 is to illustrate that recap data can provide richer semantic information. Overall, p=0.1 achieves the best performance across metrics. This means that mixing the original caption with the recaption gives the best results!

**Larger models.** We further train a larger model, DiT-L/2, for 1 epoch with a mixed ratio of $p = 0.0$, while keeping other training parameters unchanged. The model achieves an FID of 25.14 and a CLIP Score of 34.82. In Figure 6 (see appendix), we visually compare the generated results of DiT-L/2 and DiT-B/4 at $p = 0.0$. It is evident that although the quantitative scores may not show substantial improvement, as we scale up the model, there is a noticeable enhancement in the alignment between the generated images and the corresponding text, *i.e.*, this improved alignment results in higher-quality images that are able to capture and express more intricate details. These results confirm that DiT models trained on our recaption DataComp-1B exhibit robust scalability for text-to-image generative tasks.

## 7. Conclusion

This paper introduces Recap-DataComp-1B, a large-scale image dataset paired with detailed textual descriptions, generated using the LLaMA-3-powered LLaVA model. Our comprehensive analysis reveals that, compared to the original, web-crawled textual data, these generated descriptions align more accurately with their corresponding images and are more detailed. Utilizing Recap-DataComp-1B for train-

ing resulted in consistent enhancements across various models, notably CLIP, particularly in image-to-text and text-to-image retrieval tasks, and in text-to-image Diffusion models, specifically in their ability to follow more closely to user-provided text instructions. By providing this high-quality, publicly available, large-scale image-text dataset, we hope to inspire ongoing research and development that will push the boundaries of developing vision-language foundation models, more particularly in the open-source community.

## Impact Statement

This paper presents work whose goal is to advance the field of Machine Learning. There are many potential societal consequences of our work, one of it can be dataset bias. We selected Datacomp-1B as the primary image source. Although Datacomp is derived from Common Crawl and represents a snapshot of data in the public internet, it inevitably includes noisy and potentially unsafe content. The dataset's curation involves rigorous filtering, including NSFW filters on images and text, face blurring to reduce identity risks, and a take-down policy for disputed content. While these measures enhance safety, some offensive content may still persist due to the limitations of model-based filtering. We adhered to the Datacomp downloading and preprocessing procedures and only release the URLs for downloading. While using original captions may directly introduce offensive content, generating captions solely from images can also lead to unpredictable and potentially offensive model behavior, highlighting an ongoing concern that warrants further investigation.

## Acknowledge

This work is partially supported by a gift from Adobe, TPU Research Cloud (TRC) program, Google Cloud Research Credits program, AWS Cloud Credit for Research program, Edinburgh International Data Facility (EIDF) and the Data-Driven Innovation Programme at the University of Edinburgh.

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

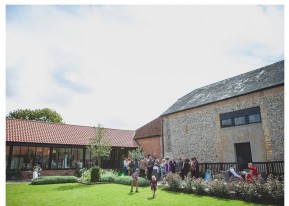

Original: 1
Recap: 4

Michelle & Karl at The Granary Barns, Newmarket 66

*A group of people are gathered outside a building with a red roof, surrounded by a lush green lawn and a variety of plants. The sky is partly cloudy, and the building appears to be a mix of modern and traditional architecture.*

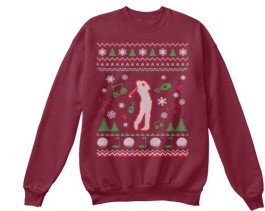

Original: 2
Recap: 3

Christmas Ugly Golf Sweater Burgundy T-Shirt Front

*A maroon sweatshirt with a festive holiday design featuring a golfer in a red sweater and white pants, holding a golf club, surrounded by snowflakes, pine trees, and musical notes. The sweatshirt has a ribbed collar and cuffs, and the design is centered on the chest.*

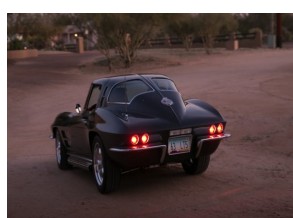

Original: 2
Recap: 4

1963 Chevrolet Corvette Resto Mod - 3 - Print Image

*A sleek black sports car with a shiny finish and a license plate that reads 'L 15' is parked on a dirt road. The car has a low profile, a pointed front, and a distinctive rear design with red taillights. The road is surrounded by a desert landscape with sparse vegetation and a wooden fence in the background.*

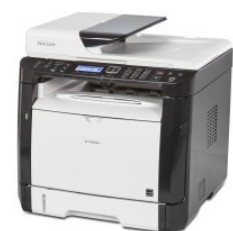

Original: 3
Recap: 4

Ricoh SP 325SFNw, Mono Laser Printer

*A white and black modern laser printer with a flatbed scanner on top, featuring a control panel and a paper tray, is displayed against a white background.*

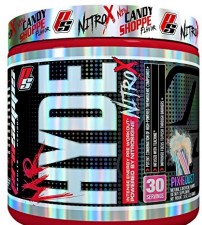

Original: 3
Recap: 3

ProSupps Mr. Hyde NitroX Pre-Workout Powder Energy Drink - Intense Sustained Energy, Pumps & Focus with Beta Alanine, Creatine & Nitrosigine - 30 True Servings

*A close-up of a red and silver canister with the brand name 'Candy Shoppe' and the product name 'NitroX'. The canister has a textured surface with a pattern of red and silver lines. The background is a gradient of red to silver, and there are no other objects or text visible.*

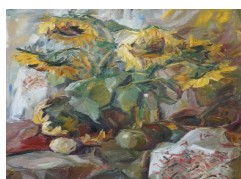

Original: 3
Recap: 3

Valerij Ivanovich Sosna. Sunflowers

*A still life painting featuring a bouquet of sunflowers in a vase, with a textured, brushstroke style that gives the image an impressionistic feel. The sunflowers are depicted in various stages of bloom, with some petals open and others closed. The vase is placed on a surface with a patterned cloth, and there are additional items such as a bowl, a small green fruit, and a red cloth. The background is a muted yellow, and the overall color palette is dominated by greens, yellows, and browns.*

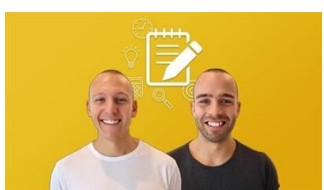

Original: 1
Recap: 3

Udemy - 7 Steps to a Perfectly Written Business Plan

*A digital illustration of two men with bald heads, one wearing a white t-shirt and the other in a dark t-shirt. They are both smiling and looking directly at the viewer. Above them is a stylized image of a notebook with a pencil on it, surrounded by various icons such as a lightbulb, a magnifying glass, and a speech bubble. The background is a solid yellow color.*

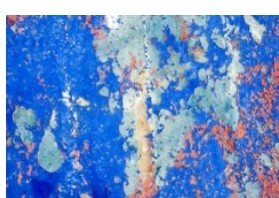

Original: 1
Recap: 4

Grunge texture Stock Image

*A close-up of a textured blue surface with orange and white speckles and patches. The surface has a rough, almost peeling appearance, with some areas looking like they are lifting off. The blue tone is dominant, but there are areas where the orange and white speckles are more prominent.*

*Figure 4.* Examples of the original caption and our recaption in our dataset. Blue box: our recpation results. Orange box: human-eval ratings.

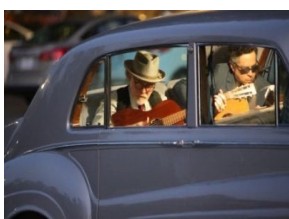

*Original: 1*
*Recap: 4*

The King \xe2\x80\x93 Starring Elvis\xe2\x80\x99s Rolls Royce \xe2\x80\x93 coming soon\xe2\x80\xa6

*A vintage car with a classic design is parked on the side of the road. The car's rear window reveals two men inside, one playing a guitar and the other holding a ukulele. The man playing the guitar is wearing a hat and glasses, while the man with the ukulele is wearing sunglasses. The car's body is a light color with a chrome bumper and a white wall tire visible in the foreground.*

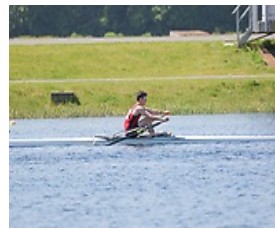

*Original: 1*
*Recap: 3*

Race: 25  Event: J15 1x  Time Trial
>\n
>\nJunior Sculling Regatta 2018
>\n
>\nTo purchase this photo, or to see pricing information for Prints and Downloads, click the blue 'Add to Cart' button at the top-right of the page.

*A person is rowing a boat on a calm lake. The individual is wearing a red top and black shorts, and is seated in the boat with oars in hand. The boat is moving through the water, leaving a trail behind it. The lake is surrounded by a grassy area, and there is a clear sky above.*

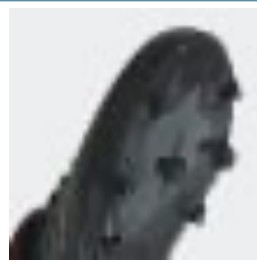

*Original: 1*
*Recap: 3*

Predator 19.3 Firm Ground Cleats Core Black / Core Black / Active Red D98003

*A close-up of a black rubber sneaker with a textured sole and a white toe cap. The sneaker is positioned against a white background, and the image is in portrait orientation.*

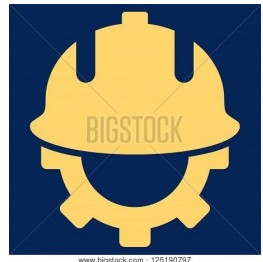

*Original: 3*
*Recap: 3*

Development Helmet vector icon. Development Helmet icon symbol. Development Helmet icon image. Development Helmet icon picture. Development Helmet pictogram. Flat yellow development helmet icon.

*A stylized yellow and blue illustration of a construction helmet with the word 'BIGSTOCK' written in a circular pattern around it. The helmet is centered on a blue background, and the image has a watermark with the text 'BIGSTOCK' and the numbers '1234567890' in the bottom right corner.*

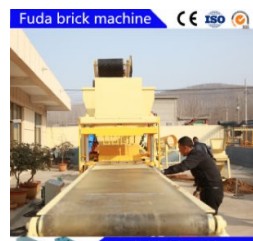

*Original: 3*
*Recap: 4*

Lego Clay Brick Molding Machine Interlocking Solid Block Machine pictures & photos

*A man is operating a yellow brick-making machine in a factory setting. The machine is large and has a conveyor belt with bricks on it. There are several potted plants around the area, and the factory has a modern look with a clear blue sky visible through the windows.*

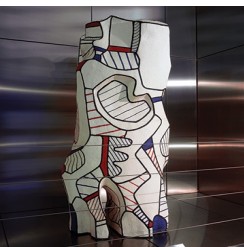

*Original: 3*
*Recap: 3*

Tour Aux Jambes by Jean Dubuffet

*A modern art sculpture with a textured surface and a variety of colors, including red, blue, and white, is displayed on a shelf in a room with a reflective floor and a large window.*

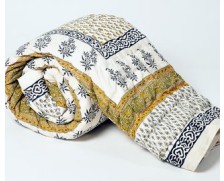

*Original: 3*
*Recap: 3*

Marwl Jaipuri Floral Print Pure Cotton Double Bed Quilt Jaipuri Razai

*A white blanket with intricate blue and gold patterns is neatly folded on a white background.*

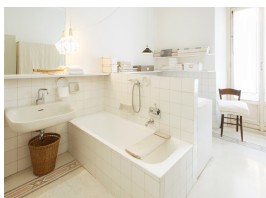

*Original: 3*
*Recap: 4*

large bathroom with tub

*A modern bathroom with white tiled walls and floor. There is a white sink with a chrome faucet to the left, a white bathtub with a chrome faucet and showerhead in the center, and a white toilet to the right. Above the sink, there is a large, round, clear glass pendant light. To the left of the sink, there is a wicker basket on the floor. The bathroom has a large window with white curtains, and there are various toiletries and towels neatly arranged on the shelves above the bathtub.*

*Figure 5.* Examples of the original caption and our recaption in our dataset. Blue box: our recpation results. Orange box: human-eval ratings.

*Table 7.* **Recap-CLIP model configurations** used in our paper.

| model | Embed dim | Vision Transformer | | | Text Transformer | | | # params (M) | | |
|---|---|---|---|---|---|---|---|---|---|---|
| | | layers | width | heads | layers | width | heads | vision | text | total |
| S/16 | 384 | 12 | 384 | 6 | 12 | 384 | 6 | 22 | 33 | 55 |
| B/16 | 512 | 12 | 768 | 12 | 12 | 512 | 8 | 86 | 53 | 141 |
| L/16 | 768 | 24 | 1024 | 16 | 12 | 768 | 12 | 303 | 109 | 414 |
| H/14 | 1024 | 32 | 1280 | 16 | 24 | 1024 | 16 | 631 | 334 | 967 |

## A. GPT-4V & Human Evaluation

We first provide two demos of detailed explanation of our GPT-4V evaluation metrics shown in Sec 4.2 of the main text. The images are shown in Figure 4, first row. Following are GPT-4V's explanations. It is clear that GPT-4V prefers Recap captions for their accuracy and detailed descriptions, capturing visual details the original captions miss. Recap excels in descriptive richness and contextual accuracy.

Example 1: Figure 1, first row, left

Recap: ***A group of people are gathered outside a building with a red roof, surrounded by a lush green lawn and a variety of plants. The sky is partly cloudy, and the building appears to be a mix of modern and traditional architecture.***

GPT-4V Rating: 4

Explanation: The given caption is mostly accurate and fluent, hence the high rating. It accurately describes a group of people gathered outside a building, which indeed has a red roof as mentioned. The lawn is green and well-maintained, matching the description of a "lush green lawn." There are also plants visible, corresponding with the mention of "a variety of plants." The sky in the image is partly cloudy, consistent with the caption. The building exhibits a blend of materials and design elements which could be seen as a mix of modern and traditional architecture especially with the modern glass elements juxtaposed with traditional stone walls.

Original: ***Michelle & Karl at The Granary Barns, Newmarket 66***

GPT-4V Rating: 1

Explanation: The image shows a wide, outdoor view with several guests apparently at an event, possibly a wedding or a gathering, by a large stone building with a red-tiled roof. There do not appear to be individuals who are prominently featured as the main subjects that can be clearly identified as Michelle and Karl. Without distinguishing features or a focus on specific individuals that could correlate to the names provided, the caption fails to correctly match the image and does not provide an accurate description of the visible content. The setting does appear to be a barnlike venue which could align with "The Granary Barns," but without clearer context or a focus on the individuals named, the overall caption quality is low for the purposes of image description.

Example 2: Figure 1, first row, right

Recap: ***A maroon sweatshirt with a festive holiday design featuring a golfer in a red sweater and white pants, holding a golf club, surrounded by snowflakes, pine trees, and musical notes. The sweatshirt has a ribbed collar and cuffs, and the design is centered on the chest.***

GPT-4V Rating: 5

Explanation: The caption accurately describes the image provided. The sweatshirt is indeed maroon and features a festive holiday design. The description of the golfer wearing a red sweater and white pants matches the figure on the sweatshirt, and the surrounding details of snowflakes, pine trees, and musical notes are all visibly present in the design. Additionally, the mention of a ribbed collar and cuffs is confirmed by the image. The design is also correctly described as being centered on the chest. The caption is fluently written and provides a detailed and accurate portrayal of the sweatshirt's design.

Original: ***Christmas Ugly Golf Sweater Burgundy T-Shirt Front***

GPT-4V Rating: 2

Explanation: The caption describes the item as a "Christmas Ugly Golf Sweater Burgundy T-Shirt," which is partially correct in context. The image shows a burgundy-colored Christmas sweater with a golf theme, which fits part of the caption's description. However, the description has inconsistencies: the garment is a sweater, not a T-shirt, and while the term "ugly" is subjective and commonly used to describe a specific style of intentionally gaudy Christmas sweaters, it's difficult to assess the accuracy of that term without an understanding of personal taste. The significant error in identifying the type of garment warrants the lower rating.

To validate GPT-4V's ratings, we conducted a human evaluation of our captions. We randomly selected 200 images from the Datacomp-1B dataset and asked human evaluators to rate both the recaptioned and original captions. To ensure objectivity, we followed a double-blind procedure, where annotators were unaware of the captions' origins. The human rating criteria are detailed below. Note that we emphasize the appearance of objects in the image as one of our key criteria, allowing us to assess the level of hallucination in the response, as suggested in the review.

1. The caption does not make sense and is completely irrelevant to the given image.

2. The caption is readable but irrelevant to the given image.

3. The caption is readable and partially related to the image, but it may contain contents that are not shown in the image.

4. The caption is fluent and related to the image, but it may contain contents that are not shown in the image.

5. The caption is fluent and exactly describes the image.

The average human evaluation results (4.3 vs. 3.1) closely align with GPT-4V's ratings, supporting our assessment of caption quality, particularly in terms of alignment.

## B. Training cost

We benchmark the inference speed of LLaVA-1.5-LLaMA3-8B on the TPU-V4 256 hardware, achieving a throughput of 382 images per second. At this rate, generating captions for the entire Recap-DataComp-1B dataset ( 1 billion images) would take approximately 29 days of continuous computation. Regarding CLIP training, training a ViT-L/16 model for two epochs ( 2.56 billion total samples) on Recap-DataComp-1B requires  1 day on TPU-V3 256 infrastructure. For DiT training, training a base-sized DiT model with a batch size of 2048 for 650K steps takes approximately 7 days using TPU-V3 256 hardware.

## C. Limitation

We selected Datacomp-1B as the primary image source to evaluate the captioning capabilities of multimodal large language models (MLLMs) like LLaVA (Liu et al., 2023b) and to explore the scalability of our synthetic captions. Although Datacomp is derived from Common Crawl and represents a snapshot of data in the public internet, it inevitably includes noisy and potentially unsafe content. The dataset's curation involves rigorous filtering, including NSFW filters on images and text, face blurring to reduce identity risks, and a take-down policy for disputed content. While these measures enhance safety, some offensive content may still persist due to the limitations of model-based filtering. We adhered to the Datacomp downloading

and preprocessing procedures and only release the URLs for downloading. While using original captions may directly introduce offensive content, generating captions solely from images can also lead to unpredictable and potentially offensive model behavior, highlighting an ongoing concern that warrants further investigation.

Second, our two-stage LLaVA training involves 558k image-text pairs from LAION (Schuhmann et al., 2022b), CC (Chang-pinyo et al., 2021), and SBU (Ordonez et al., 2011), as well as 665k instruction-following data from LLaVA-1.5 (Liu et al., 2023a) and HQ-Edit (Hui et al., 2024). Although the visual instructional data are filtered, the visual encoder and LLM are pre-trained on massive internet-sourced data, which may contain biases related to gender, race, or culture. Consequently, our caption model might inherit and perpetuate these biases. Moreover, since the model generates text probabilistically, it may produce information that is not grounded in the image, leading to potential hallucinations and impacting caption quality like other vision-language models.

Moreover, in our CLIP experiments, we found inferior performance when solely train the model on generated captions. Initially, we suspect that the model we used inherently lacks strong classification capabilities and is primarily a text-generation model. According to previous research (Zhang et al., 2024b), the training data is one of the key factors that enable MLLMs to enhance their classification ability. In our work, we focused more on improving the alignment between image and text rather than enhancing the model's classification capabilities. The training data we used did not include examples that are targeted to the classification task, which we leave as the future exploration.

Additionally, the prompts used to generate captions were designed to describe the images as accurately as possible based on the visual content, but the caption model may lack the knowledge to tell specific named entities. For example, in the second row of the teaser on the left, our captioner describe the environment and background vividly but cannot infer the specific species of the bird. Therefore, accurately describing some specific named objects within an image is one of our limitations.

Finally, due to limited computational resources and the high inference costs associated with large models, our choice of captioning model was constrained (*e.g.*, size, type).

## D. License

We have divided our licensing details into two parts: the captioner model and the generated captions. Additionally, we distribute the original Datacomp-1B images and URLs as well as original captions in accordance with their respective licenses.

- **LLaVA-Llama3 model and generated caption license:** Since our LLaVA model is based on LLaMA 3, we must strictly adhere to the original licensing terms. First, we will include a copy of the LLaMA 3 license agreement with the model. Users must comply with all terms and conditions outlined in these original licenses. For example, the generated captions are limited for further training or improving other large language models **that are not based on Llama 3 (Meta LLaMA Team, 2024)**.

- **Image-url and original caption license:** the Datacomp-1B distributes the image URL-text samples and metadata under a standard Creative Common CC-BY-4.0 license. Our improvements can be considered as a derivative work of Datacomp-1B. Therefore, we will continue to use the CC BY 4.0 license to release, retain the attribution to the original author, and clearly state that the work is based on the original DataComp-1B dataset. We will include a clear attribution such as 'This dataset is based on "DataComp-1B", created by (Gadre et al., 2023), licensed under CC BY 4.0.'

## E. DiT Qualitative Results

## F. Ablations on Model and Prompt Selections

We conduct experiments for ablating the language model, vision-language model, and the input prompt for recaptioning experiments. In detail, we choose two different language models and two vision-language model types for ablation. The detailed model recipe is below:

- Language Model: LLaMA3-8B (Meta LLaMA Team, 2024), Gemma2-27B (Gemma Team, 2024).

- Vision-Langauge Framework: LLaVA-1.5 (Liu et al., 2023a), LLaVA-NeXT (Liu et al., 2024).

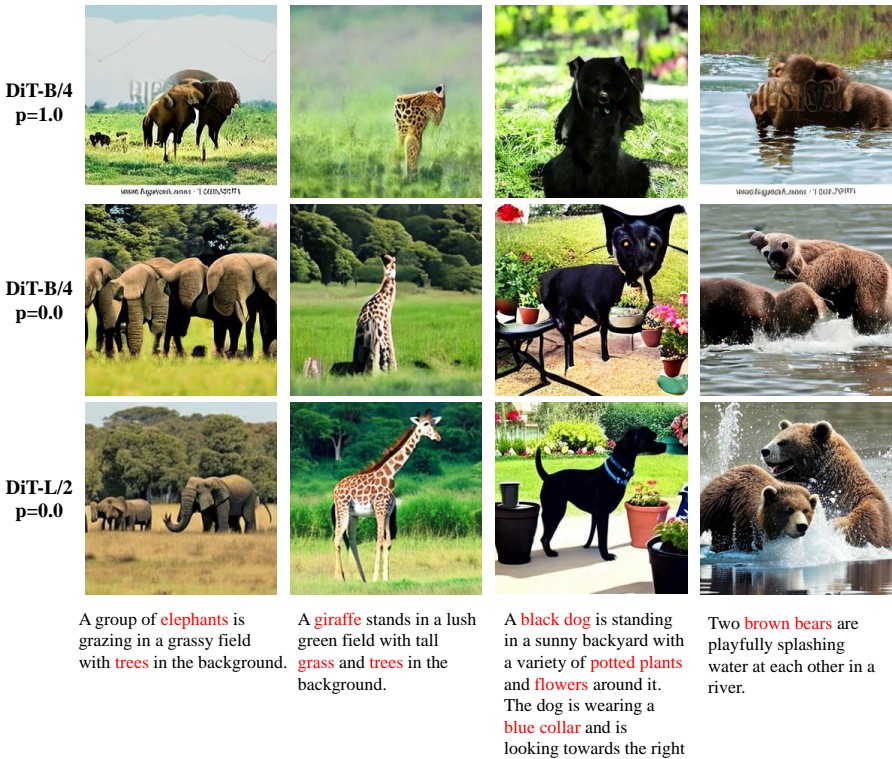

A group of elephants is grazing in a grassy field with trees in the background.

A giraffe stands in a lush green field with tall grass and trees in the background.

A black dog is standing in a sunny backyard with a variety of potted plants and flowers around it. The dog is wearing a blue collar and is looking towards the right side of the image.

Two brown bears are playfully splashing water at each other in a river.

*Figure 6.* Visual comparison of generate results from DiT-B/4 at $p = 1.0$ and $p = 0.0$, and DiT-L/2 at $p = 0.0$. We can see that decreasing $p$ enables models to comprehend texts better in image generation, and increasing model size from B/4 to L/2 improves the overall quality of generated images. We mark entities in the instruction.

As for the input prompts, we select four different ways to probe their influences to our recaption framework.

- *baseline*: The vanilla prompt input in the paper that asks the model to generate a descriptive caption of the given image.

- *w/ Brief Prompt*: We use the prompt to instruct the model to generate short and concise captions.

- *w/ Diverse Prompt*: We construct a prompt candidate pool with 11 diverse input prompts. Then we randomly sample one prompt for each captioning instruction.

- *condition on Ori. Cap.*: We use the vanilla instruction but also add the original captions from the DataComp-1B to enlighten the model with annotated knowledge.

We showcase the detailed pormpts below:

> **Original Prompt:** Please generate a detailed caption of this image. Please be as descriptive as possible.
> **Concise Prompt:** Please generate a short and clear explanation of the image. Please be as concise as possible.
> **Condition Prompt:** Please generate a detailed caption of this image. Please be as descriptive as possible based on original caption: `[ORIGINAL DATACOMP CAPTION]`.
> **Diverse Prompts Pool:**
> 1. Describe the image concisely as short as possible.
> 2. Provide a brief description of the given image.
> 3. Offer a succinct explanation of the picture presented.
> 4. Summarize the visual content of the image.
> 5. Give a short and clear explanation of the subsequent image.
> 6. Share a concise interpretation of the image provided.

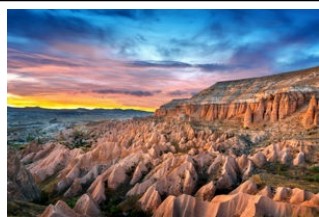 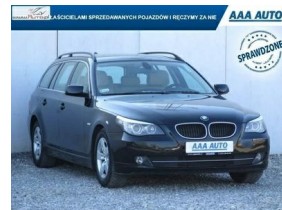

- **Orignal**: Rondreis Cappadocië & Eftalia Aytur

- **LLaVA1.5 (LLaMA3-8B):** A panoramic view of Cappadocia's unique landscape with towering rock formations and a vibrant sunset sky.

- **LLaVA1.5 (LLaMA3-8B)** *w/ Brief Prompt*: The image features a stunning landscape of a mountain range with a variety of rock formations. The sky above the mountains is filled with clouds, and the sun is setting, casting a warm glow on the scene. The colors of the sky range from blue to orange, creating a beautiful sunset.

- **LLaVA1.5 (LLaMA3-8B)** *w/ Diverse Prompt*: The image features a breathtaking landscape of a mountain range with a beautiful sunset in the background. The sky is filled with a mix of blue and orange hues, creating a stunning backdrop for the scene. The mountains are covered in rock formations, with some of them resembling a desert-like terrain. The overall atmosphere is serene and picturesque, capturing the essence of nature's beauty.

- **LLaVA1.5 (LLaMA3-8B)** *condition on Ori. Cap.*: A scenic view of Cappadocia's unique rock formations under a sky with a gradient of warm colors from sunset.

- **LLaVA-NeXT (LLaMA3-8B):** A tranquil sunset bathes the rugged landscape of a desert canyon, where the warm hues of orange and pink from the setting sun contrast with the cool blues of the evening sky. The canyon walls, sculpted from red sandstone, rise majestically, their undulating forms softened by the gentle light. The sparse vegetation at the base of the cliffs adds a touch of life to the otherwise barren landscape. This image captures the timeless beauty of a desert canyon at sunset, a moment of natural splendor where the earth and sky meet in a display of colors and contrasts.

- **LLaVA-NeXT (Gemma2-27B):** The image is a captivating landscape photograph that captures the essence of a desert environment at sunset. The rock formations are the central focus, with their varied shapes and sizes creating a sense of depth and texture. The color palette is dominated by warm tones, with the rocks ranging from light beige to deep orange, reflecting the golden hues of the setting sun. The sky above is a gradient of orange and blue, with wispy clouds adding a soft contrast. The absence of any visible human elements or structures suggests a remote, untouched landscape. This image could be used to evoke a sense of natural beauty, solitude, and the grandeur of geological formations.

- **Orignal**: używany BMW 216 Inny 1.9dm3 190KM 2008r.006km

- **LLaVA1.5 (LLaMA3-8B):** A black BMW car is parked on a gravel surface. The car has a license plate with the letters 'AAA' and 'AUTO' on it. The car is positioned in front of a wall with a logo that reads'spray' and'spray' in Polish. The car's headlights are off, and the rearview mirrors are visible.

- **LLaVA1.5 (LLaMA3-8B)** *w/ Brief Prompt*: A black BMW car is parked on a gravel surface. The car has a license plate and a sticker on the front. The license plate is visible on the front of the car, and the sticker is located on the side. The car appears to be a sedan with a sleek design.

- **LLaVA1.5 (LLaMA3-8B)** *w/ Diverse Prompt*: The image features a black BMW car parked on a gravel surface. The car is positioned in the center of the frame, with its front facing the viewer. The license plate is visible on the front of the car, and the car appears to be a luxury model. The background is not very detailed, but it seems to be a parking area or a driveway.

- **LLaVA1.5 (LLaMA3-8B)** *condition on Ori. Cap.*: A black BMW 216 with a license plate that reads 'AA AUTO' is parked on a gravel surface. The car has a'spray' logo on the side and is positioned in front of a wall with a white sticker that reads'spray'. The car has a sunroof, fog lights, and a rear spoiler.

- **LLaVA-NeXT (LLaMA3-8B):** The image is a promotional photograph of a BMW car, likely taken for the purpose of advertising or showcasing the vehicle for sale. The car's design and features, such as the headlight shape and grille, suggest it is a model from the early 2010s. The presence of the "SPRZDZONE" watermark indicates that the image may be used for online sales or promotional material, likely by an auto dealership or a car sales platform. The text "AAA AUTO" on the license plate could imply that the car is being sold by a dealership with that name or is part of a fleet for a company with that name.

- **LLaVA-NeXT (Gemma2-27B):** The image is a promotional photograph of a BMW 5 Series Touring car, which is a station wagon model. The car is presented in a showroom setting, with a clean and uncluttered background that emphasizes the vehicle's design. The car's color is a dark shade, possibly black or a very dark blue, which is a common choice for luxury vehicles as it conveys a sense of sophistication and elegance. The BMW logo on the grille is clearly visible, indicating the brand and model of the car. The presence of the "AAA Auto" logo and the "Sprzedawca" text.

*Figure 7.* Examples of image captions on DataComp-1B generated by different models with various textual prompts.

7. Present a compact description of the photo's key features.
8. Relay a brief, clear account of the picture shown.
9. Render a clear and concise summary of the photo.
10. Write a terse but informative summary of the picture.
11. Create a compact narrative representing the image presented.

We present several examples of captions generated by different recaptioning systems in Figures 7 and 8. While the original captions from DataComp-1B offer a basic understanding of the images, our recaptioning models consistently produce more descriptive, fluent, and detailed information, particularly when leveraging larger and more advanced base language models or vision-language frameworks (*e.g.*, Gemma2-27B, LLaVA-NeXT). Notably, models conditioned on the original captions from DataComp often generate even richer image descriptions, packed with additional knowledge. For instance, as shown on the right side of Figure 8, only LLaVA1.5 **with the original caption**, identifies *pancakes with salmon on the top* in the image. This highlights the benefits of integrating intricate details from the original image descriptions to enhance recaptioning performance.

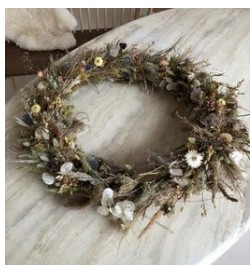

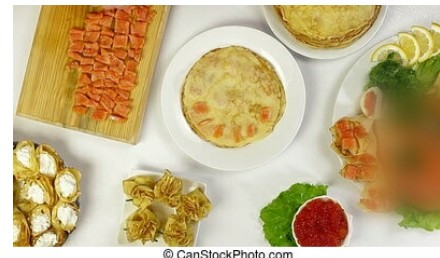

- **Orignal**: Raket & Distels Wreath of dried flowers and grasses - large

- **LLaVA1.5 (LLaMA3-8B)**: A rustic floral wreath with a mix of dried flowers, leaves, and various decorative items such as seashells and pine cones is placed on a marble surface. The wreath is circular in shape and has a natural, earthy color palette. The marble surface has a warm beige tone with visible veins, and there is a furry, light-colored object in the background.

- **LLaVA1.5 (LLaMA3-8B)** *w/ Brief Prompt*: A large, circular wreath made of various flowers and other natural elements is placed on a marble table. The wreath is adorned with a mix of flowers, leaves, and other decorative items, creating a rustic and charming appearance.

- **LLaVA1.5 (LLaMA3-8B)** *w/ Diverse Prompt*: The image features a large, circular wreath made of flowers and other natural elements, such as leaves and twigs, placed on a marble table. The wreath is quite large, occupying a significant portion of the table's surface. The table itself is white, and the wreath is positioned in the center, creating a beautiful and eye-catching display.

- **LLaVA1.5 (LLaMA3-8B)** *condition on Ori. Cap.*: A rustic wreath made of dried flowers and grasses is placed on a marble surface. The wreath is adorned with various dried flowers, leaves, and twigs, creating a natural and earthy appearance. The marble surface has a warm tone, and the wreath is positioned in the center, filling the frame.

- **LLaVA-NeXT (LLaMA3-8B)**: The image presents a natural, rustic wreath that could be used for home decor or as a seasonal centerpiece. The choice of materials suggests a focus on sustainability and the use of natural elements. The variety of colors and textures, from the softness of the flowers to the roughness of the stones, adds to the organic aesthetic. The placement of the wreath on a marble surface could indicate a setting that values both natural beauty and refined design. The image could be used to illustrate concepts such as nature-inspired decor, the beauty of imperfection, or the art of floral arrangements.

- **LLaVA-NeXT (Gemma2-27B)**: The image captures a wreath that appears to be crafted from natural materials, possibly foraged from the environment. The wreath's composition includes a variety of elements such as dried flowers, seed pods, and grasses, which are arranged in a circular pattern. The color palette is predominantly earthy tones, with the dried flowers adding pops of white and yellow. The wreath is placed on a marble surface, which provides a stark contrast to the organic materials of the wreath. The background is blurred, suggesting that the focus is solely on the wreath, which is the main subject of the image.

- **Orignal**: Pancakes With Salmon - Cooking pancakes with salmon on...

- **LLaVA1.5 (LLaMA3-8B)**: A variety of dishes are displayed on a dining table, including a round omelette with a golden-brown crust and a pattern of orange and white filling, a plate of sushi rolls, a bowl of pickled ginger, a plate of wasabi, and a small dish of soy sauce.

- **LLaVA1.5 (LLaMA3-8B)** *w/ Brief Prompt*: The image displays a variety of food items on a dining table. There are two plates of food, one with a quiche and the other with a pizza. In addition, there are bowls containing salad and a plate with sushi. The table also has a plate of shrimp and a bowl of sauce. The food is arranged in a visually appealing manner, with a focus on the quiche and pizza.

- **LLaVA1.5 (LLaMA3-8B)** *w/ Diverse Prompt*: The image displays a table filled with a variety of food items, including a large plate of sushi, a plate of eggs, and a plate of meat. There are also bowls of food, such as a bowl of sauce and a bowl of salad. The table is set with multiple plates, bowls, and a spoon, indicating a meal is being prepared or served. The variety of dishes suggests a diverse and delicious meal is being enjoyed.

- **LLaVA1.5 (LLaMA3-8B)** *condition on Ori. Cap.*: A variety of dishes are displayed on a table, including a plate of pancakes with salmon on top, a plate of sushi, a plate of dumplings, a plate of sushi rolls, a plate of sushi nigiri, a plate of sushi sashimi, a plate of sushi maki, a plate of sushi nigiri, a plate of sushi sashimi, a plate of sushi maki, a plate of sushi nigiri, a plate of sushi sashimi, a plate of sushi maki, a plate of sushi nigiri, a plate of sushi sashimi, a plate of sushi maki.

- **LLaVA-NeXT (LLaMA3-8B)**: The image is a culinary display that showcases a variety of dishes, likely intended for a meal or a buffet. The arrangement is symmetrical, with each dish placed in a way that creates a balanced composition. The use of white plates and a white tablecloth suggests a clean and minimalist aesthetic, which is often associated with fine dining or a special occasion. The presence of both sweet and savory items indicates a variety of options, catering to different tastes. The image could be used for culinary presentation, menu design, or as a visual reference for food styling. It's important to note that the image is a representation of food.

- **LLaVA-NeXT (Gemma2-27B)**: The image is a collection of various food items, likely from a buffet or a shared meal setting. The dishes include a savory pie, a quiche, and a dessert, each with its own distinct characteristics. The pie has a golden-brown crust, suggesting it's been baked to a crisp, while the quiche is creamy and golden, indicating a rich egg and cheese filling. The dessert is a light and airy pastry, likely filled with a sweet cream or custard. The presentation is simple yet elegant, with the dishes arranged in a way that highlights the variety of textures and colors.

*Figure 8.* Examples of image captions on DataComp-1B generated by different models with various textual prompts.

