# OpenReview forum: "What If We Recaption Billions of Web Images with LLaMA-3?"
_ICML.cc/2025/Conference — ICML 2025 poster_

### Official Review · Reviewer_fVXe · 2025-03-12

**Overall Recommendation:** 3

**Summary:**

The authors finetune a better Llava-1.5 model using a more advanced Llama model. The authors then recaption the DataComp-1B dataset and show promising results in terms of ImageNet zero-shot accuracy and text-to-image retrieval benchmarks. The authors finally use the CLIP score to show that the recaptioned-caption is better correlated with the image than the initial web-scraped caption.

**Claims And Evidence:**

The claims are well supported.

**Essential References Not Discussed:**

N/A

**Experimental Designs Or Analyses:**

As explained in the "Methods And Evaluation Criteria" section, I don't think the zero-shot and cross modal retrieval benchmarks are the best metrics to showcase the superiority of the new recaptioned dataset. (Tables 2,3,4,5).

I do think Table 6's experiment on text-to-image generation is a step in the right direction. However, Table 6 is less comprehensive and not the focus of the paper.

**Methods And Evaluation Criteria:**

Strengths:
- The dataset itself is very useful to researchers. For example, it could be used to train better open-sourced CLIP models.

Weaknesses:
- There are no algorithmic contributions. The authors use off-the-shelf methods.
- In my opinion, there is some serious misalignment between what the authors are trying to say about their dataset and what type of results the authors show in the tables. The authors show that their generated caption is more descriptive; consequently a CLIP model trained on the recaptioned dataset does not do well in zero-shot classification (see Table 3 p=0.0). Even for text-to-image retrieval and image-to-text retrieval tasks, the authors have to set p to 0.8 (a larger portion of original captions) to see some recall improvement. This paper could benefit from more complex benchmarks that test the model's understanding of the image (such as image captioning and VQA).
- I know DataComp is derived from LAION, which was taken down a year ago. Is this data still publicly available?

**Other Comments Or Suggestions:**

Page 5 line 227: "We can observe ..." This sentence refers to some numbers that are not cited. Please cite the corresponding table.

**Other Strengths And Weaknesses:**

See above.

**Questions For Authors:**

Is this dataset derived from the LAION dataset with safety fixes (https://laion.ai/blog/relaion-5b/)? If not, the impact of this work is limited, because you can't release the dataset.

**Relation To Broader Scientific Literature:**

The proposed novel dataset is a significant contribution to the multimodal literature.

**Theoretical Claims:**

N/A

---

> ### Author Rebuttal · Authors · 2025-04-01
>
> ### **Q1: No algorithmic contributions.**
>
> **A1:**  While we don’t introduce new recaptioning algorithms, we highlight two core contributions:
> Recap-DataComp-1B, the first fully open, billion-scale image-text dataset with synthetic captions generated using LLaMA-3. Prior works [1,2] are proprietary and closed-source. Scaling recaptioning with advanced LLMs at this scale is unprecedented and enables reproducible, large-scale research in multimodal learning.
>
>
> This dataset supports the first public large-scale evaluation of CLIP and T2I diffusion models trained on high-quality synthetic captions. Our experiments show notable gains in cross-modal tasks, long-context understanding, and image generation, making Recap-DataComp-1B a valuable resource for the community.
>
>
> [1] Improving Image Generation with Better Captions. 2023
> [2] A Picture is Worth a Thousand Words: Principled Recaptioning Improves Image. 2023
>
> ---
>
> ### **Q2: Misalignment between their dataset’s advantage and results.**
>
> **A2:** Our main goal is to improve noisy web captions by aligning them better with images. Hence, we focus on tasks like cross-modal retrieval that directly measure image-caption alignment. We also include human and GPT-based evaluations (Sec. 4.2 and Appendix A) to assess caption quality. In classification, training only on synthetic captions degrades CLIP performance. However, mixing just 10% of original captions substantially recovers accuracy, confirming that original captions still play a key role in avoiding data collapse. In contrast, T2I diffusion benefits from detailed, synthetic captions. Setting the mixing ratio to p=0.1 (i.e., 90% recaptions) leads to better generation results since richer captions produce better text latents.
>
> ---
>
> ### **Q3:  More complex benchmarks (such as image captioning and VQA). Table 6 is less comprehensive.**
>
> **A3:**  Thank you for the suggestion. Our goal is to provide a high-quality, large-scale dataset for both understanding and generation tasks. We conducted large-scale CLIP and T2I experiments to demonstrate its effectiveness and potential. Table 6 is limited by the high computational cost—for instance, training CLIP ViT-B/16 for 2 epochs only takes approximately 0.5 days, while DiT-B/4 requires around 7 days for a single epoch on TPU-v3-256. These constraints restricted the number of benchmarks we could include. Furthermore, the current results already demonstrate our dataset’s effectiveness, and we will leave further evaluations to future work.
>
>
>  Following your advice, we adopted the LLaVA framework to evaluate more challenging tasks such as VQA and image captioning. We replaced its pretraining data (558K LAION/CC3M/SBU) with 558K Recap-DataComp-1B samples. Results show consistent gains, including +0.6% across four VQA tasks and +34 points on MME shown in the following table:
>
> | Pre-train Dataset                | Tunable Module | Training Steps | TextVQA | MME        | VQA-V2 | MM-Vet   |
> |----------------------------------|----------------|----------------|---------|------------|--------|----------|
> | LLaVA-LCS-558K                   | Projector      | 2K             | 59.1    | 1489/277 | 77.9   | 34.4     |
> | Recap-DataComp-1B (Recaption Only) | Projector      | 2K             | 60.1 | 1523/260 | 78.6 | 35.1 |
>
>
> Lastly, we further evaluated the CLIP visual encoder trained with our mix-training strategy on our dataset. Our LLaVA with ViT trained only on synthetic captions (p=0) shows only a slight drop compared to training on the original data (p=1) — an average of 1.2% across four VQA benchmarks (TextVQA, GQA, MM-Vet, SEED). Notably, at p=0.8, LLaVA achieves the best overall performance, matching the p=1 model and surpassing it on MME by 35 absolute points.
>
> | Model | Mix ratio |  IN-1K | Text VQA | GQA  | MME  | MM-Vet | SEED |
> |-------|-----------|-----------------|----------|------|------|--------|------|
> | B/16  | p=0       | 33.8            | 50.0     | 58.9 | 1335 | 25.0   | 62.8 |
> | B/16  | p=0.8     | 69.8            | 52.0     | 60.2 | 1417 | 25.6   | 64.2 |
> | B/16  | p=1       | 70.5            | 51.8     | 60.0 | 1382 | 25.6   | 63.9 |
>
>
> These results demonstrate that by evaluating more diverse and challenging tasks, our proposed dataset not only significantly enhances cross-modal retrieval performance but also reveals promising
>
> ---
>
>
>
> ### **Q4: Is this data DataComp still publicly available? Safety fixes of LAION.**
>
> **A4:** Yes, DataComp-1B remains public and is not derived from LAION-5B but from a different Common Crawl version. The  original dataset is accessible at Huggingface Dataset. More importantly, it applied strict NSFW filtering, and we discussed safety considerations in the Appendix. We believe releasing our recaptioned dataset poses no major safety risks.
>
> ---
>
> **Other**:  on page 5, line 227, we cite the GPT-4V rating. We will clarify this in the revision.

---

### Official Review · Reviewer_xErf · 2025-03-14

**Overall Recommendation:** 3

**Summary:**

This paper explores the impact of improving textual descriptions for large-scale web-crawled image-text datasets using LLaMA-3. The authors propose a recaptioning pipeline that fine-tunes a LLaMA-3-8B-powered LLaVA-1.5 model and applies it to ∼1.3 billion images from the DataComp-1B dataset. The resulting dataset, Recap-DataComp-1B, enhances training for vision-language models. Experiments demonstrate that for discriminative models like CLIP, the recaptioned dataset improves zero-shot performance on four cross-modal retrieval tasks. For generative models like Diffusion Transformers, the refined captions enable better alignment with text instructions, particularly in handling complex queries. In general, this paper is well organized.

**Claims And Evidence:**

yes

**Essential References Not Discussed:**

NA

**Experimental Designs Or Analyses:**

yes

**Methods And Evaluation Criteria:**

yes

**Other Comments Or Suggestions:**

NA

**Other Strengths And Weaknesses:**

Strengths:
+ The paper contributes an open-source, large-scale recaptioning pipeline, fostering community-wide research in vision-language model training.
+ The study demonstrates quantitative gains for both discriminative and generative models, enhancing retrieval accuracy and text-to-image generation quality.
+ By systematically refining web-crawled captions, the proposed method mitigates inherent noise, leading to more semantically rich and effective training data.

Weaknesses:
- Recaptioning has been explored and verified effective in prior work, like DALL-E 3 [a] and [b]. Compared with existing work, this work seems not to bring impressive discovery, insights , or conclusions.

- The novelty of this work is weak and unclear. I understand this work may focus on empirical verification, but an empirical work also needs to provide some novel ideas or experience to contribute the development of the research community.

- Performance improvement of using regenerated captions over that with raw captions on some tasks (such as the results in Tab 1 and Tab 4) is marginal.

- Weaker performance w/ re-caption on ImageNet-1K in Tab 2 can be observed. Is there any other task that re-caption may be harmful? More analysis may be helpful, which should not weaken the quality of this work, but lets the community pay more attention to the potential disadvantages or risks of recaption.


[a] Improving Image Generation with Better Captions. 2023
[b] A Picture is Worth a Thousand Words: Principled Recaptioning Improves Image. 2023

**Questions For Authors:**

See above

**Relation To Broader Scientific Literature:**

NA

**Theoretical Claims:**

yes

---

> ### Author Rebuttal · Authors · 2025-04-01
>
> ### **Q1: Novel ideas,  insights, and conclusions.**
>
> **A1:** Thank you for raising concerns about novelty. Our primary focus is on developing **Recap-DataComp-1B**, which we believe to be **the first publicly available image-text dataset at the billion-scale generated by LLaMA-3**. Unlike previous works [1,2] that remain proprietary—both in terms of captioning models and datasets—our approach is fully open and demonstrably scalable to the billion level. We believe this openness and scale set a new milestone in multimodal research.
>
> While the idea of “recaptioning” is not entirely new, we have yet to see it executed at this scale using advanced large language models. By creating recaptions on such a massive level, our dataset enables the first public, large-scale experiments into training models like CLIP and T2I diffusion with high-quality synthetic captions. In our view, **Recap-DataComp-1B** offers a novel and significant contribution to the field, with considerable potential to advance future multimodal research.
>
> [1] Improving Image Generation with Better Captions. 2023
> [2] A Picture is Worth a Thousand Words: Principled Recaptioning Improves Image. 2023
>
> ---
>
> ### **Q2: Marginal improvement on some tasks.**
>
> **A2:**  Thank you for the question. As shown in Table 1, the LLaVA-3 **8B** model offers slightly better performance than the **13B** model while being significantly faster. We selected it to balance captioning quality and efficiency.
> In Table 4, Recap-CLIP clearly outperforms the previous DataComp-1B baseline trained on the same samples, highlighting the improved quality of our dataset. While our results slightly exceed those of SigLIP, it’s important to note that SigLIP requires four times more training samples  on a private dataset, which is five times larger—making our gains particularly noteworthy.
>
> ---
>
>
> ### **Q3: Weaker performance on ImageNet-1K. Are more tasks harmful?**
>
> **A3:**  Thanks for your constructive suggestion. We observe that training exclusively with synthetic captions degrades CLIP performance. However, introducing a small portion of original captions (e.g., 10%) effectively recovers performance, indicating that original captions remain crucial in preventing data collapse, as noted in prior work [1]. Additionally, we clarify that while recent works like LaCLIP [2] and VeCLIP [3] demonstrate synthetic captions can enhance CLIP training, to our knowledge, no prior work has trained models exclusively on synthetic captions. Our study represents the first public attempt, shedding light on the challenges of this approach.
>
>
> Second, we further evaluate the visual representations learned from these models through linear probing and full fine-tuning on ImageNet-1K. While the CLIP ViT trained only on synthetic captions underperforms in zero-shot classification, We observe that lightweight tuning (e.g., linear probing or fine-tuning) greatly narrows the performance gap seen in zero-shot settings. Moreover, models trained on mixed captions perform on par with those trained on original data, striking a favorable balance between classification and retrieval performance.
>
> | Model | Mix ratio | Zero-shot | Linear Prob | Fully FT |
> |-------|-----------|-----------------|-------------|----------|
> | B/16  | p=0       | 33.8            | 76.1        | 83.6     |
> | B/16  | p=0.8     | 69.8            | 80.3        | 84.2     |
> | B/16  | p=1       | 70.5            | 80.4        | 84.1     |
>
> Following your advice, we also evaluated additional tasks to analyze the impact of recaption following the original DataComp paper across 22 datasets (VTAB contains 13, IN-1K-Shift contains 6, and Retrieval contains 3), shown in the table below. We found that purely synthetic captions negatively impacted zero-shot performance. However, our mix-training strategy still achieves superior performance in VTAB and retrieval tasks.
>
> | Model | Mix ratio | IN-1K | VTAB | IN-1K-Shift | Retrieval |
> |-------|-----------|-----------------|------|-------------|-----------|
> | B/16  | p=0       | 33.8            | 38.0 | 33.3        | 42.2      |
> | B/16  | p=0.8     | 69.8            | 57.5 | 55.5        | 56.6      |
> | B/16  | p=1       | 70.5            | 56.8 | 55.6        | 54.9      |
>
>
> These results indicate that only synthetic captions may negatively impact zero-shot classification performance. Our mixed-training strategy effectively balances these competing factors. We acknowledge that further investigation into potential drawbacks is valuable. Given the demonstrated scalability and high-quality alignment of our proposed dataset, we believe it provides a strong baseline for future research.
>
> ---
>
> [1] Seddik M E A, et al. How bad is training on synthetic data? a statistical analysis of language model collapse[J]. arXiv, 2024.
> [2] Fan, Lijie, et al. "Improving CLIP training with language rewrites." NeurIPS. 2024.
> [3] Lai, Zhengfeng, et al. "VeCLIP: Improving CLIP training via visual-enriched captions." ECCV.  2024.

---

### Official Review · Reviewer_ogfc · 2025-03-14

**Overall Recommendation:** 4

**Summary:**

This paper explores the impact of recaptioning web-crawled image-text pairs using LLaMA-3. The authors identify that web-crawled datasets (like DataComp-1B) suffer from image-text misalignment and low-quality textual descriptions. Their approach is straightforward: they fine-tune a LLaMA-3-8B powered LLaVA-1.5 model and use it to generate detailed captions for approximately 1.3 billion images from DataComp-1B, creating a new dataset called Recap-DataComp-1B.

The paper demonstrates that these enhanced captions are longer, more diverse, and better aligned with their corresponding images. Through extensive experiments, they show that vision-language models trained on this enhanced dataset exhibit significant improvements: CLIP models achieve an average 3.1% boost in zero-shot cross-modal retrieval performance when trained on a mix of original and recaptioned data, while text-to-image Diffusion Transformers show better alignment with text instructions and produce higher-quality images (demonstrated by improved FID scores and CLIP scores).

**Claims And Evidence:**

Most claims in the paper are well-supported by clear and convincing evidence:

1. The claim that web-crawled image-text pairs are inherently noisy and misaligned is effectively demonstrated through visual examples in Figure 1, comparing original captions with recaptions.
2. The performance improvement of their LLaMA-3-powered LLaVA model over other LLaVA variants is adequately supported by comparisons on MMMU and MM-Vet benchmarks in Table 1.
3. The claim that Recap-DataComp-1B contains higher-quality captions is validated through:
   - Word and length distributions (Figures 1 & 3)
   - GPT-4V evaluations showing an average rating increase from 3.71 to 4.14
   - LongCLIP scores demonstrating better semantic alignment (89.91 vs. 10.09)
4. The benefits of training vision-language models on recaptioned data are thoroughly evidenced in Tables 2-5 for CLIP models and Table 6 for DiT models.

The only claim that could use stronger evidence is the assertion that recaptioning enhances classification performance. The results in Table 3 show that using purely recaptioned data (p=0.0) actually hurts ImageNet classification. While the authors acknowledge this and pick a good compromise at p=0.8 for later ablations, more analysis of this limitation would strengthen the paper.

**Essential References Not Discussed:**

Not noticed.

**Experimental Designs Or Analyses:**

The experimental designs are sound and well-executed:

1. CLIP training experiments (Section 5) are comprehensive, exploring mixing ratios, text encoder sizes, and evaluating on multiple benchmarks. The decision to use p=0.8 as a compromise between classification and retrieval performance is well-justified.
2. Text-to-image experiments (Section 6) properly evaluate how different mixing ratios affect generation quality, with both quantitative metrics and visual examples.
3. Caption quality analyses (Section 4) use multiple complementary methods to assess semantic alignment and descriptiveness.

A significant limitation, however, is that while the paper extensively discusses how recaptioned data affects downstream models, all experiments are conducted using a single recaptioning model configuration. The paper lacks systematic quantitative analysis of how different settings of the caption model itself (base LLM selection, image encoder variations, training data choices, or whether conditioning on original captions) affect caption quality and subsequent downstream performance. Although the authors provide qualitative examples in the supplementary materials (Figures 7 and 8), quantitative ablation studies on the captioning pipeline itself would significantly strengthen the work and provide insights into which components most strongly influence the quality of the recaptioned dataset.

**Methods And Evaluation Criteria:**

The methods and evaluation criteria are appropriate and comprehensive:

1. The evaluation of caption quality uses a multi-faceted approach:
   - Automatic metrics (CLIP, LongCLIP)
   - Human-in-the-loop evaluation (GPT-4V)
   - Qualitative examples and word distribution analysis
2. For CLIP evaluation, the authors perform extensive ablations:
   - Testing different mixing ratios between original and recaptioned data
   - Experimenting with various text encoder sizes
   - Evaluating on standard benchmarks (ImageNet-1K, COCO, Flickr30K)
   - Additional evaluations on challenging datasets (Urban1K, VG-Attribution)
3. For DiT evaluation, they use standard metrics (FID, CLIP scores) as well as GPT-4V scoring, which provides a more holistic assessment of generation quality.

The experimental design includes appropriate baselines and ablations, making the conclusions reliable.

**Other Comments Or Suggestions:**

1. It would be valuable to analyze potential biases in the generated captions, as they may inherit biases present in the training data of LLaMA-3 and LLaVA.
2. Showing some failure cases where the recaptioning model produces inaccurate descriptions would help understand the limitations of the approach.
3. The appendix mentions experiments with different prompting strategies and conditioning on original captions. These findings could be highlighted more prominently in the main paper.
4. Releasing smaller, curated subsets of the recaptioned dataset would benefit researchers with limited computational resources.

**Other Strengths And Weaknesses:**

Strengths:

1. The paper tackles an important problem (improving web-crawled image-text data) at an unprecedented scale (1.3 billion images).
2. The approach is practical and accessible, using open-source models rather than closed APIs.
3. The comprehensive evaluation across multiple model types (CLIP and DiT) and tasks demonstrates the broad applicability of the approach.
4. The analysis of optimal mixing ratios provides practical insights for future researchers.

Weaknesses:

1. Limited exploration of prompt engineering: The paper uses a single prompt template for recaptioning. Testing multiple prompting strategies might yield more diverse captions.
2. The degradation in classification performance when using only recaptioned data (p=0.0) is noted but not extensively analyzed.
3. The human evaluation is limited to GPT-4V ratings rather than traditional human evaluation, though the large-scale nature of the dataset makes this a practical choice.

**Questions For Authors:**

1. You found that training with purely recaptioned data (p=0.0) significantly hurts classification performance on ImageNet-1K. Do you have insights into why this happens, and have you explored techniques to mitigate this issue while preserving the benefits for retrieval tasks?
2. In the appendix, you mention experimenting with conditioning the recaptioning model on original captions. Did you evaluate training vision-language models directly on these condition-based recaptions? Would this approach help with the classification performance drop observed with pure recaptions?

**Relation To Broader Scientific Literature:**

The paper is well-positioned within the broader scientific literature:

1. It builds upon foundational work on vision-language models like CLIP (Radford et al., 2021) and text-to-image models (DiT, Diffusion Transformers).
2. It clearly differentiates from previous recaptioning approaches like ShareGPT4V (Chen et al., 2023) and LaCLIP (Fan et al., 2024) by emphasizing:
   - The use of open-source models (LLaMA-3) rather than closed APIs
   - The billion-scale application, which is significantly larger than previous efforts
   - The comprehensive evaluation across both discriminative and generative models
3. The authors acknowledge the high costs associated with using API-based approaches like GPT-4V for billion-scale datasets, highlighting the practical significance of their approach.

**Theoretical Claims:**

The paper is primarily empirical and does not make significant theoretical claims requiring formal proofs.

---

> ### Author Rebuttal · Authors · 2025-04-01
>
> ### **Q1: Analysis of this limitation, insights, and technology to migrate classification performance issues.**
>
> **A1:**  Following your valuable suggestion,  we analyze the classification drop with synthetic captions and identify three likely causes: (1) CLIP's difficulty learning from long, rich captions [1], (2) captioners’ poor performance on classification tasks [2], and (3) limited evaluation of zero-shot only. We additionally evaluate performance on the standard ImageNet-1K classification task using both linear probing and full fine-tuning. We observe that introducing lightweight tuning significantly reduces the initial performance gap seen in zero-shot settings, indicating that models trained solely on synthetic captions still learn strong visual representations.
>
> | Model | Mix ratio | Zero-shot  | Linear Prob | Fully FT |
> |-------|-----------|-----------------|-------------|----------|
> | B/16  | p=0       | 33.8            | 76.1        | 83.6     |
> | B/16  | p=0.8     | 69.8           | 80.3        | 84.2     |
> | B/16  | p=1       | 70.5            | 80.4        | 84.1     |
>
>
> Regarding potential strategies, mixed training already achieves a preferable balance of classification and retrieval, and recent work [3] suggests that advanced CLIP training methods can further exploit synthetic captions.
>
> ---
>
>
>
> ### **Q2: Caption model and quantitative results.**
>
> **A2:** Thanks for bringing up the concern about ablation on caption models.  We evaluate four caption models on a 30M subset and the results are shown in the following Table.  We find that LLaVA-1.5-LLaMA3-8B consistently yields the best downstream CLIP performance. Larger models like LLaVA-NeXT-Gemma2-27B are significantly slower—requiring 201 days to recaption the full 1 billion samples on the same infrastructure—and also underperform on retrieval tasks.  Thus, we chose LLaVA-1.5-LLaMA3-8B for re-captioning the full dataset, balancing quality and efficiency.
>
> | Model | Caption Model | Caption Speed | Mix Ratio | IN1K | Flickr T2I | Flickr I2T | COCO T2I | COCO I2T |
> |-------|---------------------------|---------------|-----------|------|------------|------------|----------|----------|
> | L/16  | -                         | -             | 1.0       | 66.1 | 48.6       | 65.3       | 30.2     | 41.7     |
> | L/16  | LLaVA-1.5-LLaMA3-8B       | 382 img/s     | 0.6       | 67.5 | 61.1   |77.8   | 39.5 | 54.0 |
> | L/16  | LLaVA-next-Gemma2-27B     | 54 img/s      | 0.6       | 67.1 | 58.9       | 74.6       | 37.0     | 51.8     |
>
>
> ---
>
>
> ### **Q3:  Prompt engineering and quantitative results.**
>
> **A3:**  We ablate prompt engineering on a 30M subset and found prompt choice influences downstream tasks shown in the following table:
>
> | Model | Prompt Type      | Mix Ratio | IN1K  | Flickr T2I | Flickr I2T | COCO T2I | COCO I2T |
> |-------|------------------|-----------|-------|------------|------------|----------|----------|
> | L/16  | -                | 1.0       | 66.1  | 48.6       | 65.3       | 30.2     | 41.7     |
> | L/16  | Original-prompt  | 0.6       | 67.5  | 61.1       | 77.8       | 39.5     | 54.0     |
> | L/16  | Concise-prompt   | 0.6       | 67.8  | 61.5       | 80.6       | 40.5     | 57.1     |
> | L/16  | Diverse-prompt   | 0.6       | 68.2 | 63.7 | 81.4  | 42.3 | 57.3 |
> | L/16  | Condition-prompt | 0.6       | 68.2 | 62.0     | 78.7       | 40.1     | 55.5     |
>
> Our findings show that even simple prompts, such as the *Original* setup, already significantly enhance retrieval performance at this scale. As the first to release a billion-caption dataset, these promising results motivate us to further expand and refine our evaluations in future work.
>
> ---
>
>
> ### **Q4: Limited Human Evaluation.**
>
> **A4:**  We present human evaluation results in Appendix A and Figures 4 and 5, which align closely with GPT-4V assessments. We will further emphasize this consistency in future revisions.
>
>
> ---
>
> ### **Q5: Potential biases and failure cases in the generated captions.**
>
> **A5:**  We acknowledge the presence of biases and failure cases in generated captions inherited from the training data of LLaMA-3 and LLaVA. Addressing and mitigating these biases will be a key focus of our future work. Regarding presenting potential failure cases, one example of a current shortcoming is the pipeline’s inability to correctly identify certain objects—for instance, the “Western Kingbird” shown in Figure 1. We will include a more detailed analysis of failure cases in the next version.
>
>
> ---
>
>
>
> ### **Q6: Smaller, curated subset.**
>
> **A6:**  We will release the ablation subset to support further research.
>
> ---
>
> [1] Zhang, Beichen, et al. "Long-CLIP: Unlocking the long-text capability of CLIP." ECCV, 2024.
> [2] Zhang, Yuhui, et al. "Why are visually-grounded language models bad at image classification?” arXiv, 2024.
> [3] Liu, Yanqing, et al. "CLIPS: An Enhanced CLIP Framework for Learning with Synthetic Captions." arXiv,  2024.

---

> > ### Comment · Reviewer_ogfc · 2025-04-02
> >
> > I have read the author's rebuttal and the other reviews. The authors have adequately addressed the concerns raised, including providing additional analysis and results.
> >
> > While the core recaptioning technique isn't novel, the scale, use of open models, and the public release of this large dataset represent a significant and valuable contribution to the community. The extensive evaluations and insights derived from this work are solid.
> >
> > Therefore, I maintain my recommendation for 4:Accept.

---

> > > ### Author Response · Authors · 2025-04-04
> > >
> > > Dear Reviewer ogfc
> > >
> > > Thank you for recognizing and acknowledging the value of this work — we truly appreciate it! Please feel free to let us know if you have any other questions.
> > >
> > > Thanks
> > > Authors

---

### Official Review · Reviewer_Manh · 2025-03-14

**Overall Recommendation:** 2

**Summary:**

The paper introduces Recap-Datacomp-1B a large scale image-text data, where the text is "synthetically generated" using large multimodal models. The authors observed that compared to original web-crawled data, models trained on this generated synthetic texts are performing better for multimodal retrieval and  text-to-image generation tasks.

**Claims And Evidence:**

Experimental results corroborate the claim that the recap dataset is better for training multimodal models.

**Essential References Not Discussed:**

I can see that others have trained LLaVa-LLaMa3 models earlier: https://huggingface.co/Intel/llava-llama-3-8b
Is there any reason to train the model again?

**Ethical Review Concerns:**

Thank you for providing the ethical statements about how the limitation of model-based filtering might have let some of the unsafe content to be used for training and specifying the copyright details.

**Experimental Designs Or Analyses:**

Experiments are rigorous,

**Methods And Evaluation Criteria:**

Results are shown on standard tasks using acceptable metrics.

**Other Comments Or Suggestions:**

Although it's a new dataset with improved fine-grained image-text data, I find its technical innovation is missing.

**Other Strengths And Weaknesses:**

It's understood that the fine-grained aligned multimodal data is essential for learning good image-text models as per our current definition of goodness. That has been the motivation behind creating larger and larger datasets. As observed by the authors, the improvement especially about fine-grained textual input following in text-to-image diffusion models is notable. I'd argue that the research question on the other hand should be about how to incorporate common sense, or reducing dependencies on large scale annotations.

**Questions For Authors:**

I encourage the authors to provide training time. Currently there has been no mention of the complexity associated with creating this Recap dataset; training models using this newly generated recap dataset.

**Relation To Broader Scientific Literature:**

It's understood that the fine-grained aligned multimodal data is essential for learning good image-text models as per our current definition of goodness. That has been the motivation behind creating larger and larger datasets. As observed by the authors, the improvement especially about fine-grained textual input following in text-to-image diffusion models is notable. I'd argue that the research question on the other hand should be about how to incorporate common sense, or reducing dependencies on large scale annotations.

**Theoretical Claims:**

NA

---

> ### Author Rebuttal · Authors · 2025-04-01
>
> Thank you for recognizing that our dataset significantly enhances performance in both multimodal retrieval and text-to-image generation tasks. We appreciate your insights into how advanced MLLMs can reduce reliance on costly human annotations and enable greater scalability.
> Our Recap data relies solely on LLaVA-based annotation, which **inherently incorporates commonsense knowledge** from a human perspective — since LLaVA is an aligned model trained on human preference data. In other words, our Recap approach not only reduces dependence on manual annotation but also opens up promising directions for future research.
> Our experiments further show that mixed-caption training — integrating both CLIP and T2I methods — offers a robust way to incorporate **external** commonsense knowledge, leading to consistent performance improvement.
>
> In essence, our dataset serves as a valuable baseline and starting point for exploring these innovative strategies. We plan to expand on the broader impact of this work in the upcoming version.
>
> ---
>
> ### **Q1: Why not use earlier LLaVa-LLaMa3 models?**
>
> **A1:** Thank you for bringing up this work. First, we trained our own version to maintain full control over the training process while adhering to open-research practices and a broader licensing framework. Second, our initial investigation revealed that the original LLaVA-1.5 data consistently started its outputs with "This image is a …". To address this issue, we integrated a high-quality dataset (HQ-Edit) into our training process, which alleviated the problem. We will include this essential reference in our next version.
>
> ---
>
>
> ### **Q2: Technical innovation.**
>
> **A2:** We would like to stress that our primary focus is on creating **Recap-DataComp-1B**, which, to the best of our knowledge, is **the first publicly available image-text dataset with synthetic captions scaled to the billion level using LLaMA-3**. We believe this represents a novel and significant contribution to the multimodal research community. While the concept of recaptioning is not new, scaling it to this magnitude with advanced LLMs has not been seen before. More importantly, this large-scale dataset enables the first public, extensive, and fair investigations into training CLIP and T2I diffusion models with high-quality synthetic captions. For example, our results comprehensively demonstrate that Recap-DataComp-1B significantly enhances cross-modal tasks, long-context understanding, and text-to-image generation. Based on this evidence, we believe that Recap-DataComp-1B is a novel and important contribution to the community, with the strong potential to provide significant benefits to future multimodal research.
>
>
> ---
>
>
> ### **Q3: Complexity of creating and training models of our dataset.**
>
> **A3:** We benchmark the inference speed of LLaVA-1.5-LLaMA3-8B on the TPU-V4 256 hardware, achieving a throughput of 382 images per second. At this rate, generating captions for the entire Recap-DataComp-1B dataset (~1 billion images) would take approximately 29 days of continuous computation.
>
> Regarding CLIP training, training a ViT-L/16 model for two epochs (~2.56 billion total samples) on Recap-DataComp-1B requires ~ 1 day on TPU-V3 256 infrastructure. For DiT training, training a base-sized DiT model with a batch size of 2048 for 650K steps takes approximately 7 days using TPU-V3 256 hardware.
>
> We will provide comprehensive details about computational complexity and runtime metrics in the next revision.

---

### Decision · Program_Chairs · 2025-05-01

**Decision:**

Accept (poster)

**Comment:**

This paper examines the effects of re-captioning at scale web-crawled image-text pairs using LLaMA-3. The authors highlight that datasets like DataComp-1B, which are obtained through web crawling, often experience issues with image-text misalignment and low-quality textual descriptions. To address this, they propose using a fine-tuned LLaVA-1.5 with LLaMA-3-8B to generate detailed captions for approximately 1.3 billion images from DataComp-1B, resulting in a new dataset called Recap-DataComp-1B.

While reviewers had concerns about the impact of the recaption data on the ImageNet zero-shot classification tasks, the authors addressed this point during rebuttal by providing more empirical evidence (linear/finetuning protocol, VTAB evaluation...).

Overall, all reviewers highlighted the limited novelty of the approach, as recaptionning has been explored in prior work. However, reviewers also emphasized that releasing a large-scale recaption dataset would be very useful for the community and to foster more progress in multimodal research. I agree with the reviewer's assessment and support acceptance.